# Global Geometry of Multichannel Sparse Blind Deconvolution on the Sphere

**Yanjun Li**
CSL and Department of ECE
University of Illinois
Urbana-Champaign
yli145@illinois.edu

**Yoram Bresler**
CSL and Department of ECE
University of Illinois
Urbana-Champaign
ybresler@illinois.edu

## Abstract

Multichannel blind deconvolution is the problem of recovering an unknown signal $f$ and multiple unknown channels $x_i$ from convolutional measurements $y_i = x_i \circledast f$ ($i = 1, 2, \ldots, N$). We consider the case where the $x_i$'s are sparse, and convolution with $f$ is invertible. Our nonconvex optimization formulation solves for a filter $h$ on the unit sphere that produces sparse output $y_i \circledast h$. Under some technical assumptions, we show that all local minima of the objective function correspond to the inverse filter of $f$ up to an inherent sign and shift ambiguity, and all saddle points have strictly negative curvatures. This geometric structure allows successful recovery of $f$ and $x_i$ using a simple manifold gradient descent algorithm with random initialization. Our theoretical findings are complemented by numerical experiments, which demonstrate superior performance of the proposed approach over the previous methods.

## 1   Introduction

Blind deconvolution, which aims to recover unknown vectors $x$ and $f$ from their convolution $y = x \circledast f$, has been extensively studied, especially in the context of image deblurring [1, 2, 3]. Recently, algorithms with theoretical guarantees have been proposed for single channel blind deconvolution [4, 5, 6, 7, 8, 9, 10]. In order for the problem to be well-posed, these previous methods assume that *both* $x$ and $f$ are constrained, to either reside in a known subspace or be sparse over a known dictionary [11, 12]. However, these methods cannot be applied if $f$ (or $x$) is unconstrained, or does not have a subspace or sparsity structure.

In many applications in communications [13], imaging [14], and computer vision [15], convolutional measurements $y_i = x_i \circledast f$ are taken between a single signal (resp. filter) $f$ and multiple filters (resp. signals) $\{x_i\}_{i=1}^N$. We call such problems multichannel blind deconvolution (MBD). Importantly, in this multichannel setting, one can assume that only $\{x_i\}_{i=1}^N$ are structured, and $f$ is unconstrained. While there has been abundant work on single channel blind deconvolution (with both $f$ and $x$ constrained), research on MBD (with $f$ unconstrained) is relatively limited. Traditional MBD works assumed that the channels $x_i$'s are FIR filters [16, 17, 18] or IIR filters [19], and proposed to solve MBD using subspace methods. The problem is generally ill-conditioned, and the recovery using the subspace methods is highly sensitive to noise [20].

In this paper, while retaining the unconstrained form of $f$, we consider a different structure of the multiple channels $\{x_i\}_{i=1}^N$: sparsity. The resulting problem is termed multichannel sparse blind deconvolution (MSBD). The sparsity structure arises in many real-world applications.

**Opportunistic underwater acoustics:** Underwater acoustic channels are sparse in nature [21]. Estimating such sparse channels with an array of receivers using opportunistic sources (e.g., shipping noise) involves a blind deconvolution problem with multiple unknown sparse channels [22, 23].

**Reflection seismology:** Thanks to the layered earth structure, reflectivity in seismic signals is sparse. It is of great interest to simultaneous recover the filter (also known as the wavelet), and seismic reflectivity along the multiple propagation paths between the source and the geophones [24].

**Functional MRI:** Neural activity signals are composed of brief spikes and are considered sparse. However, observations via functional magnetic resonance imaging (fMRI) are distorted by convolving with the hemodynamic response function. A blind deconvolution procedure can reveal the underlying neural activity [25].

**Super-resolution fluorescence microscopy:** In super-resolution fluorescence microscopic imaging, photoswitchable probes are activated stochastically to create multiple sparse images and allow microscopy of nanoscale cellular structures [26, 27]. One can further improve the resolution via a computational deconvolution approach, which mitigates the effect of the point spread function (PSF) of the microscope [28]. It is sometimes difficult to obtain the PSF (e.g., due to unknown aberrations), and one needs to jointly estimate the microscopic images and the PSF [29].

Previous approaches to MSBD have provided efficient iterative algorithms to compute maximum likelihood (ML) estimates of parametric models of the channels $\{x_i\}_{i=1}^N$ [23], or maximum a posteriori (MAP) estimates in various Bayesian frameworks [24, 15]. However, these algorithms usually do not have theoretical guarantees. Recently, guaranteed algorithms for MSBD have been developed. Wang and Chi [30] proposed a convex formulation of MSBD based on $\ell_1$ minimization. Li et al. [31] solved a nonconvex formulation using projected gradient descent, and proposed an initialization algorithm to compute a sufficiently good starting point. However, the theoretical guarantees of these algorithms require restrictive assumptions (e.g., $f$ has one dominant entry that is significantly larger than other entries [30], or $f$ has an approximately flat spectrum [31]).

We would like to emphasize that, while earlier papers on MBD [16, 17, 18, 19] consider a linear convolution model, more recent guaranteed methods for MSBD [30, 31] consider a circular convolution model. By zero padding the signal and the filter, one can rewrite a linear convolution as a circular convolution. In practice, circular convolution is often used to approximate a linear convolution when the filter has a compact support or decays fast [32], and the signal has finite length or satisfies a circular boundary condition [1]. The accelerated computation of circular convolution via the fast Fourier transform (FFT) is especially beneficial in 2D or 3D applications [1, 29]. Multichannel blind deconvolution with a circular convolution model is also related to blind gain and phase calibration with Fourier measurements [33, 34, 35, 36, 37].

In this paper, we consider MSBD with circular convolution. In addition to the sparsity prior on the channels $\{x_i\}_{i=1}^N$, we impose, without loss of generality, the constraint that $f$ has unit $\ell_2$ norm, i.e., $f$ is on the unit sphere. (This eliminates the scaling ambiguity inherent in the MBD problem.) We show that our sparsity promoting objective function has a nice geometric landscape on the the unit sphere: **(S1)** all local minima correspond to signed shifted versions of the desired solution, and **(S2)** the objective function is strongly convex in neighborhoods of the local minima, and has strictly negative curvature directions in neighborhoods of local maxima and saddle points. Similar geometric analysis has been conducted for dictionary learning [38], phase retrieval [39], and single channel sparse blind deconvolution [10]. Recently, Mei et al. [40] analyzed the geometric structure of the empirical risk of a class of machine learning problems (e.g., nonconvex binary classification, robust regression, and Gaussian mixture model). This paper is the first such analysis for MSBD.

Although our analysis of global geometry shares a similar roadmap with previous works [10, 38, 39, 40], much of our theoretical analysis is tailored for MSBD. For example, our partition of the unit sphere into three regions (of strong convexity, negative curvature, and large gradient, respectively) is carefully crafted for our objective function, and is closely related to our error bound. We leverage tools that are commonly used in related works, such as concentration inequalities and union bounds, to prove the geometric properties. However, our bounds are derived specifically for MSBD, under new assumptions. For example, the single channel sparse blind deconvolution [10] with sparse $x$, requires $f$ to have compact support. In contrast, in this work on MSBD, other than invertibility, we make no assumptions on $f$.

Properties **(S1)** and **(S2)** allow simple manifold optimization algorithms to find the ground truth in the nonconvex formulation. Unlike the second order methods in previous works [41, 39], we take advantage of recent advances in the analysis of first-order methods [42, 43], and prove that a simple manifold gradient descent algorithm, with random initialization and a fixed step size, can accurately recover a signed shifted version of the ground truth in polynomial time almost surely. This is the first guaranteed algorithm for MSBD that does *not* rely on restrictive assumptions on $f$ or $\{x_i\}_{i=1}^N$.

Recently, many optimization methods have been shown to escape saddle points of objective functions with benign landscapes, e.g., gradient descent [44, 45], stochastic gradient descent [46], perturbed gradient descent [47], Natasha [48, 49], and FastCubic [50]. Similarly, optimization methods over Riemannian manifolds that can escape saddle points include manifold gradient descent [43], the trust region method [41, 39], and the negative curvature method [51]. Our main result shows that these algorithms can be applied to MSBD thanks to the favorable geometric structure of our objective.

## 2    MSBD on the Sphere

### 2.1    Problem Statement

In MSBD, the measurements $y_1, y_2, \ldots, y_N \in \mathbb{R}^n$ are the circular convolutions of unknown sparse vectors $x_1, x_2, \ldots, x_N \in \mathbb{R}^n$ and an unknown vector $f \in \mathbb{R}^n$, i.e., $y_i = x_i \circledast f$. In this paper, we solve for $\{x_i\}_{i=1}^n$ and $f$ from $\{y_i\}_{i=1}^N$. One can rewrite the measurement as $Y = C_f X$, where $C_f$ represents the circulant matrix whose first column is $f$, and $Y = [y_1, y_2, \ldots, y_N]$ and $X = [x_1, x_2, \ldots, x_N]$ are $n \times N$ matrices. Without structures, one can solve the problem by choosing any invertible circulant matrix $C_f$ and compute $X = C_f^{-1} Y$. The fact that $X$ is sparse narrows down the search space.

Even with sparsity, the problem suffers from inherent scale and shift ambiguities. Suppose $\mathcal{S}_j : \mathbb{R}^n \to \mathbb{R}^n$ denotes a circular shift by $j$ positions, i.e., $\mathcal{S}_j(x)_{(k)} = x_{(k-j)}$ for $j, k \in [n]$. Here we use $x_{(j)}$ to denote the $j$-th entry of $x \in \mathbb{R}^n$ (treated as modulo $n$). Note that we have $y_i = x_i \circledast f = (\alpha \mathcal{S}_j(x_i)) \circledast (\alpha^{-1} \mathcal{S}_{-j}(f))$ for every nonzero $\alpha \in \mathbb{R}$ and $j \in [n]$. Therefore, MSBD has equivalent solutions generated by scaling and circularly shifting $\{x_i\}_{i=1}^n$ and $f$.

Throughout this paper, we assume that the circular convolution with the signal $f$ is invertible, i.e., there exists a filter $g$ such that $f \circledast g = e_1$ (the first standard basis vector). Equivalently, $C_f$ is an invertible matrix, and the discrete Fourier transform (DFT) of $f$ is nonzero everywhere. Since $y_i \circledast g = x_i \circledast f \circledast g = x_i$, one can find $g$ by solving the following optimization problem:

$$\text{(P0)} \quad \min_{h \in \mathbb{R}^n} \ \frac{1}{N} \sum_{i=1}^N \|C_{y_i} h\|_0, \quad \text{s.t. } h \neq 0.$$

The constraint eliminates the trivial solution that is 0. If the solution to MSBD is unique up to the aforementioned ambiguities, then the only minimizers of (P0) are $h = \alpha \mathcal{S}_j g$ ($\alpha \neq 0, j \in [n]$).

### 2.2    Smooth Formulation

Minimizing the non-smooth $\ell_0$ "norm" is usually challenging. Instead, one can choose a smooth surrogate function for sparsity. It is well-known that minimizing the $\ell_1$ norm can lead to sparse solutions [52]. An intuitive explanation is that the sparse points on the unit $\ell_2$ sphere (which we call unit sphere from now on) have the smallest $\ell_1$ norm. As demonstrated in Figure 1, these sparse points also have the largest $\ell_4$ norm. Therefore, maximizing the $\ell_4$ norm, a surrogate for the "spikiness" [53] of a vector, is akin to minimizing its sparsity.

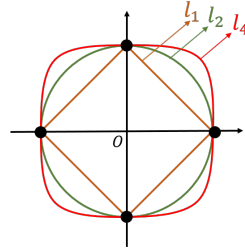

Figure 1: Unit $\ell_1$, $\ell_2$, and $\ell_4$ spheres in 2-D.

Here, we make two observations: (1) one can eliminate the scaling ambiguity by restricting $h$ to the unit sphere $S^{n-1}$; (2) sparse recovery can be achieved by maximizing $\|\cdot\|_4^4$. Based on these observations, we adopt the following optimization problem:

$$\text{(P1)} \quad \min_{h \in \mathbb{R}^n} \ -\frac{1}{4N} \sum_{i=1}^N \|C_{y_i} R h\|_4^4, \quad \text{s.t. } \|h\| = 1.$$

The matrix $R := (\frac{1}{\theta nN}\sum_{i=1}^{N} C_{y_i}^{\top} C_{y_i})^{-1/2} \in \mathbb{R}^{n \times n}$ is a preconditioner, where $\theta$ is a parameter that is proportional to the sparsity level of $\{x_i\}_{i=1}^{N}$. In Section 3, under specific probabilistic assumptions on $\{x_i\}_{i=1}^{N}$, we explain how the preconditioner $R$ works.

Problem (P1) can be solved using first-order or second-order optimization methods over Riemannian manifolds. The main result of this paper provides a geometric view of the objective function over the sphere $S^{n-1}$ (see Figure 3). We show that some off-the-shelf optimization methods can be used to obtain a solution $\hat{h}$ close to a scaled and circularly shifted version of the ground truth. Specifically, $\hat{h}$ satisfies $C_f R\hat{h} \approx \pm e_j$ for some $j \in [n]$, i.e., $R\hat{h}$ is approximately a signed and shifted version of the inverse of $f$. Given solution $\hat{h}$ to (P1), one can recover $f$ and $x_i$ ($i = 1, 2, \ldots, N$) as follows:

$$\hat{f} = \mathcal{F}^{-1}\big[\mathcal{F}(R\hat{h})^{\odot-1}\big], \qquad \hat{x}_i = C_{y_i} R\hat{h}. \tag{1}$$

Here, we use $x^{\odot-1}$ to denote the entrywise inverse of $x$.

## 3  Global Geometric View

In this paper, we assume that $\{x_i\}_{i=1}^{N}$ are random sparse vectors, and $f$ is invertible:

(A1) The channels $\{x_i\}_{i=1}^{N}$ follow a Bernoulli-Rademacher model. More precisely, $x_{i(j)} = A_{ij}B_{ij}$ are independent random variables, $B_{ij}$'s follow a Bernoulli distribution $\text{Ber}(\theta)$, and $A_{ij}$'s follow a Rademacher distribution (taking values $1$ and $-1$, each with probability $1/2$).

(A2) The circular convolution with the signal $f$ is invertible. We use $\kappa$ to denote the condition number of $f$, which is defined as $\kappa := \frac{\max_j |(\mathcal{F}f)_{(j)}|}{\min_k |(\mathcal{F}f)_{(k)}|} = \frac{\sigma_1(C_f)}{\sigma_n(C_f)}$, i.e., the ratio of the largest and smallest magnitudes of the DFT of $f$.

The Bernoulli-Rademacher model is a special case of the Bernoulli–sub-Gaussian models. The derivation in this paper can be repeated for other sub-Gaussian nonzero entries, with different tail bounds. We use the Rademacher distribution for simplicity.

Let $\phi(x) = -\frac{1}{4}\|x\|_4^4$. Its gradient and Hessian are defined by $\nabla_\phi(x)_{(j)} = -x_j^3$, and $H_\phi(x)_{(jk)} = -3x_j^2\delta_{jk}$. (We use $H_{(jk)}$ to denote the entry of $H \in \mathbb{R}^{n \times n}$ in the $j$-th row and $k$-th column, and use $\delta_{jk}$ to denote the Kronecker delta.) Then the objective function in (P1) is $L(h) = \frac{1}{N}\sum_{i=1}^{N} \phi(C_{y_i} Rh)$, where $R = (\frac{1}{\theta nN}\sum_{i=1}^{N} C_{y_i}^{\top} C_{y_i})^{-1/2}$. The gradient and Hessian are $\nabla_L(h) = \frac{1}{N}\sum_{i=1}^{N} R^{\top} C_{y_i}^{\top} \nabla_\phi(C_{y_i} Rh)$, and $H_L(h) = \frac{1}{N}\sum_{i=1}^{N} R^{\top} C_{y_i}^{\top} H_\phi(C_{y_i} Rh) C_{y_i} R$. Since $L(h)$ is to be minimized over $S^{n-1}$, we use optimization methods over Riemannian manifolds [54]. To this end, we define the tangent space at $h \in S^{n-1}$ as $\{z \in \mathbb{R}^n : z \perp h\}$ (see Figure 2). We study the Riemannian gradient and Riemannian Hessian of $L(h)$ (gradient and Hessian along the tangent space

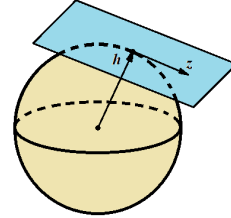

Figure 2: A demonstration of the tangent space of $S^{n-1}$ at $h$, the origin of which is translated to $h$. The Riemannian gradient and Riemannian Hessian are defined on tangent spaces.

at $h \in S^{n-1}$): $\widehat{\nabla}_L(h) = P_{h^\perp}\nabla_L(h)$, and $\widehat{H}_L(h) = P_{h^\perp} H_L(h) P_{h^\perp} - \langle\nabla_L(h), h\rangle P_{h^\perp}$, where $P_{h^\perp} = I - hh^{\top}$ is the projection onto the tangent space at $h$. We refer the readers to [54] for a more comprehensive discussion of these concepts.

The toy example in Figure 3 demonstrates the geometric structure of the objective function on $S^{n-1}$. (As shown later, the quantity $\mathbb{E}L''(h)$ is, up to an unimportant rotation of the coordinate system, a good approximation to $L(h)$.) The local minima correspond to signed shifted versions of the ground truth (Figure 3(a)). The Riemannian gradient is zero at stationary points, including local minima, saddle points, and local maxima of the objective function when restricted to the sphere $S^{n-1}$. (Figure 3(b)). The Riemannian Hessian is positive definite in the neighborhoods of local minima, and has at least one strictly negative eigenvalue in the neighborhoods of local maxima and saddle points (Figure 3(c)). We say that a stationary point is a "strict saddle point" if the Riemannian Hessian has at least one strictly negative eigenvalue. Our main result Theorem 3.1 formalizes the observation that $L(h)$ only has two types of stationary points: (1) local minima, which are close to signed shifted versions

of the ground truth, and (2) strict saddle points. Please refer to the supplementary result for the full proof.

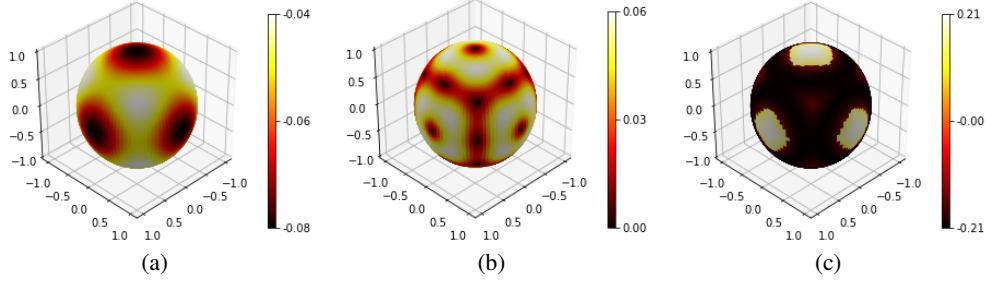

Figure 3: Geometric structure of the objective function over the sphere. For $n = 3$, we plot the following quantities on the sphere $S^2$: (a) $\mathbb{E}L''(h)$, (b) $\|\mathbb{E}\widehat{\nabla}_{L''}(h)\|$, and (c) $\min_{z \perp h, \|z\|=1} z^\top \mathbb{E}\widehat{H}_{L''}(h)z$.

**Theorem 3.1.** *Suppose Assumptions (A1) and (A2) are satisfied, and the Bernoulli probability satisfies $\frac{1}{n} \leq \theta < \frac{1}{3}$. Let $\kappa$ be the condition number of $f$, and let $\rho < 10^{-3}$ be a small tolerance constant. There exist constants $c_1, c_1', c_2, c_2' > 0$ (depending only on $\theta$), such that: if $N > \max\{\frac{c_1 n^9}{\rho^4}\log\frac{n}{\rho}, \frac{c_2\kappa^8 n^8}{\rho^4}\log n\}$, then with probability at least $1 - n^{-c_1'} - n^{-c_2'}$, every local minimum $h^*$ in (P1) is close to a signed shifted version of the ground truth. I.e., for some $j \in [n]$: $\|C_f Rh^* \pm e_j\| \leq 2\kappa\sqrt{\rho}$. Moreover, one can partition $S^{n-1}$ into three sets $\mathcal{H}_1$, $\mathcal{H}_2$, and $\mathcal{H}_3$, which, for some $c(n, \theta, \rho) > 0$, satisfy:*

- *$L(h)$ is strongly convex in $\mathcal{H}_1$, i.e., $\min_{\substack{z:\|z\|=1 \\ z \perp h}} z^\top \widehat{H}_L(h)z \geq c(n, \theta, \rho) > 0$.*

- *$L(h)$ has negative curvature in $\mathcal{H}_2$, i.e., $\min_{\substack{z:\|z\|=1 \\ z \perp h}} z^\top \widehat{H}_L(h)z \leq -c(n, \theta, \rho) < 0$.*

- *$L(h)$ has a descent direction in $\mathcal{H}_3$, i.e., $\|\widehat{\nabla}_L(h)\| \geq c(n, \theta, \rho) > 0$.*

*Clearly, all the stationary points of $L(h)$ on $S^{n-1}$ belong to $\mathcal{H}_1$ or $\mathcal{H}_2$. The stationary points in $\mathcal{H}_1$ are local minima, and the stationary points in $\mathcal{H}_2$ are strict saddle points.*

*Proof Sketch.* Note that $R = (\frac{1}{\theta n N}\sum_{i=1}^N C_{y_i}^\top C_{y_i})^{-1/2}$ asymptotically converges to $(C_f^\top C_f)^{-1/2}$ as $N$ increases. Therefore, $L(h)$ can be approximated by $L'(h) = \frac{1}{N}\sum_{i=1}^N \phi(C_{y_i}(C_f^\top C_f)^{-1/2}h) = \frac{1}{N}\sum_{i=1}^N \phi(C_{x_i}C_f(C_f^\top C_f)^{-1/2}h)$. Since $C_f(C_f^\top C_f)^{-1/2}$ is an orthogonal matrix, one can study the objective function $L''(h') = \frac{1}{N}\sum_{i=1}^N \phi(C_{x_i}h')$ with $h' = C_f(C_f^\top C_f)^{-1/2}h$, which is a rotated version of $L'(h)$ on the sphere. Our analysis consists of three parts:

(1) *Geometric structure of $\mathbb{E}L''$:* We first bound $\min_{z:\|z\|=1, z \perp h} z^\top \mathbb{E}\widehat{H}_{L''}(h)z$, which is strictly positive near its local minima, and strictly negative near all other stationary points (the strict saddle points). At the same time, at all other points on $S^{n-1}$ (the points further away from stationary points), the Riemannian gradient of $\mathbb{E}L''$ is bounded away from zero.

(2) *Deviation of $L''$ (or its rotated version $L'$) from $\mathbb{E}L''$:* We bound $\|\widehat{\nabla}_{L''}(h) - \mathbb{E}\widehat{\nabla}_{L''}(h)\|$ and $\|\widehat{H}_{L''}(h) - \mathbb{E}\widehat{H}_{L''}(h)\|$ using the matrix Bernstein inequality and union bounds.

(3) *Difference between $L$ and $L'$:* We bound $\|\widehat{\nabla}_L(h) - \widehat{\nabla}_{L'}(h)\|$ and $\|\widehat{H}_L(h) - \widehat{H}_{L'}(h)\|$ using the matrix Bernstein inequality and Lipschitz continuity of $\widehat{\nabla}_L(h)$ and $\widehat{H}_L(h)$.

Theorem 3.1 follows by combining the above results. □

# 4 Optimization Method

Recently, first-order methods have been shown to escape strict saddle points with random initialization [44, 45]. In this paper, we use the manifold gradient descent algorithm studied by Lee et al. [43]. One can initialize the algorithm with a random $h^{(0)}$, and use the following iterative update:

$$h^{(t+1)} = \mathcal{A}(h^{(t)}) := P_{S^{n-1}}\big(h^{(t)} - \gamma \widehat{\nabla}_L(h^{(t)})\big). \tag{2}$$

Each iteration takes a Riemannian gradient descent step in the tangent space, and does a retraction by normalizing the iterate (projecting onto $S^{n-1}$). Using the geometric structure introduced in Section 3, and some technical results in [42, 43], the following result gives a theoretical guarantee for manifold gradient descent for our formulation of MSBD: convergence to an accurate estimate (up to the inherent sign and shift ambiguity) of the true solution.

**Theorem 4.1.** *Suppose that the geometric structure in Theorem 3.1 is satisfied. If manifold gradient descent (2) is initialized with a random $h^{(0)}$ drawn from a uniform distribution on $S^{n-1}$, and the step size is chosen as $\gamma = \frac{1}{128n^3}$, then (2) converges to a local minimum of $L(h)$ on $S^{n-1}$ almost surely. It particular, after at most $T = \frac{4096n^8}{\theta^2(1-3\theta)^2\rho^4}$ iterations, $h^{(T)} \in \mathcal{H}_1$. Moreover, for some $j \in [n]$*

$$\|C_f R h^{(T)} \pm e_j\| \le 2\kappa\sqrt{\rho}.$$

**Corollary 4.2.** *If the conditions of Theorem 4.1 are satisfied, then the recovered $\hat{f}$ and $\hat{x}_i$ in (1), computed using the output of manifold gradient descent $\hat{h} = h^{(T)}$, satisfy (for some $j \in [n]$):*

$$\frac{\|\hat{x}_i \pm \mathcal{S}_j(x_i)\|}{\|x_i\|} \le 2\kappa\sqrt{\rho n}, \qquad \frac{\|\hat{f} \pm S_{-j}(f)\|}{\|f\|} \le \frac{2\kappa\sqrt{\rho n}}{1 - 2\kappa\sqrt{\rho n}}.$$

Theorem 4.1 and Corollary 4.2 show that, with a random initialization and a fixed step size, manifold gradient descent outputs, in polynomial time, a solution that is close to a signed and shifted version of the ground truth. We prove these results in the supplementary material.

# 5 Numerical Experiments

## 5.1 Deconvolution with Synthetic Data

In this section, we examine the empirical performance of manifold gradient descent (2) in solving MSBD (P1). We synthesize $\{x_i\}_{i=1}^N$ following the Bernoulli-Rademacher model, and synthesize $f$ following a Gaussian distribution $N(\mathbf{0}_{n\times 1}, I_n)$. In all experiments, we run manifold gradient descent for $T = 100$ iterations, with a fixed step size of $\gamma = 0.1$.

Recall that the desired $h$ is a signed shifted version of the ground truth, i.e., $C_f R h = \pm e_j$ ($j \in [n]$). Therefore, to evaluate the accuracy of the output $h^{(T)}$, we compute $C_f R h^{(T)}$ with the true $f$, and declare successful recovery if $\|C_f R h^{(T)}\|_\infty / \|C_f R h^{(T)}\| > 0.95$, or equivalently, if $\max_{j\in[n]} |\cos \angle (C_f R h^{(T)}, e_j)| > 0.95$. We compute the success rate based on 100 Monte Carlo instances. In a typical successful instance, $h^{(t)}$ converges to an accurate estimate of the ground truth after about 50 iterations (as shown by the error and accuracy plots in Figure 4(d) and 4(h)).

In the first experiment, we fix $\theta = 0.1$ (sparsity level, mean of the Bernoulli distribution), and run experiments with $n = 32, 64, \ldots, 256$ and $N = 32, 64, \ldots, 256$ (see Figure 4(a)). In the second experiment, we fix $n = 256$, and run experiments with $\theta = 0.02, 0.04, \ldots, 0.16$ and $N = 32, 64, \ldots, 256$ (see Figure 4(b)). The empirical phase transitions suggest that, for sparsity level relatively small (e.g., $\theta < 0.16$), there exist a constant $c > 0$ such that manifold gradient descent can recover a signed shifted version of the ground truth with $N \ge cn\theta$.

In the third experiment, we examine the phase transition with respect to $N$ and the condition number $\kappa$ of $f$, which is the ratio of the largest and smallest magnitudes of its DFT. To synthesize $f$ with specific $\kappa$, we generate the DFT $\tilde{f}$ of $f$ that is random with the following distribution: (1) The DFT $\tilde{f}$ is symmetric, i.e., $\tilde{f}_{(j)} = \tilde{f}_{(n+2-j)}$, so that $f$ is real. (2) The phase of $\tilde{f}_{(j)}$ follows a uniform distribution on $[0, 2\pi)$, except for the phases of $\tilde{f}_{(1)}$ and $\tilde{f}_{(n/2+1)}$ (if $n$ is even), which are always 0 for symmetry. (3) The gains of $\tilde{f}$ follows a uniform distribution on $[1, \kappa]$. We fix $n = 256$ and

$\theta = 0.1$, and run experiments with $\kappa = 1, 2, 4, \ldots, 128$ and $N = 32, 64, \ldots, 256$ (see Figure 4(c)). The phase transition suggests that the number $N$ for successful empirical recovery is not sensitive to the condition number $\kappa$.

Manifold gradient descent is robust against noise. We repeat the above experiments with noisy measurements: $y_i = x_i \circledast f + \sigma \varepsilon_i$, where $\varepsilon_i$ follows a Gaussian distribution $N(\mathbf{0}_{n \times 1}, I_n)$. The phase transitions for $\sigma = 0.1\sqrt{n\theta}$ (SNR $\approx 20\,\mathrm{dB}$) are shown in Figure 4(e), 4(f), and 4(g). For a reasonable noise level, the number $N$ of noisy measurements we need to accurately recover a signed shifted version of the ground truth is roughly the same as with noiseless measurements.

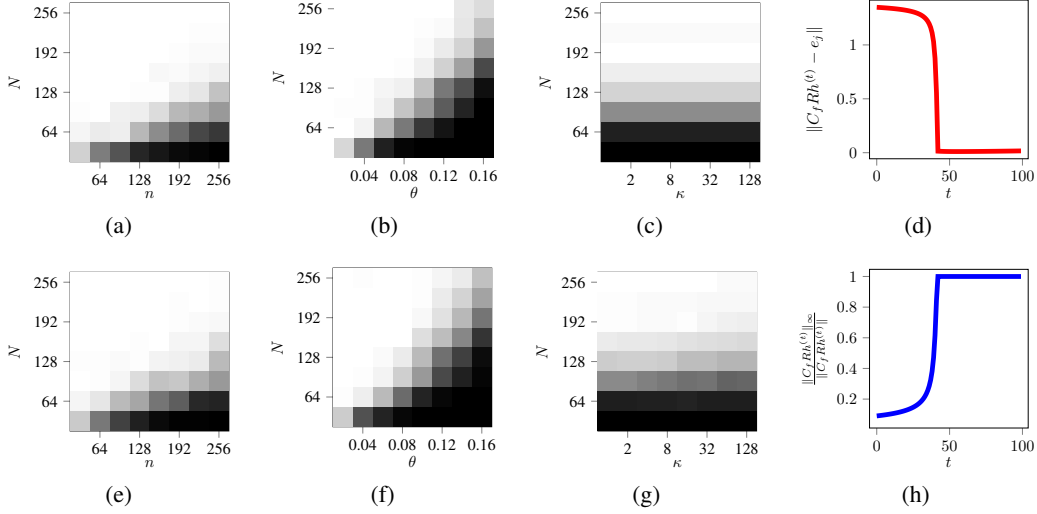

Figure 4: Empirical phase transition (grayscale values represent success rates). The first column shows the phase transitions of $N$ versus $n$. The second column shows the phase transitions of $N$ versus $\theta$. The third column shows the phase transitions of $N$ versus $\kappa$. (a) - (c) are the results for the noiseless case. (e) - (g) are the results for SNR $\approx 20\,\mathrm{dB}$. (d) and (h) show the error $\|C_f R h^{(t)} - e_j\|$ and the accuracy $\|C_f R h^{(t)}\|_\infty / \|C_f R h^{(t)}\|$ as functions of the iteration number $t$, respectively.

## 5.2 Blind Gain and Phase Calibration

In this section, we consider the blind calibration problem [31]. Suppose that a sensing system takes Fourier measurements of unknown signals, with sensors that have unknown gains and phases, i.e., $\tilde{y}_i = \mathrm{diag}(\tilde{f})\mathcal{F} x_i$, where $x_i$ are the targeted unknown sparse signals, $\mathcal{F}$ is the DFT matrix, and the entries of $\tilde{f}$ represent the unknown gains and phases. In sensor array processing [55], the supports of $x_i$'s are identical, and represent the directions of arrival of incoming sources. The simultaneous recovery of $\tilde{f}$ and $x_i$'s is equivalent to MSBD in the frequency domain.

Clearly, Assumption (A1) is not satisfied in this case. For complex $f, x_i \in \mathbb{C}^n$, we solve:

$$\min_{h \in \mathbb{C}^n} \frac{1}{N} \sum_{i=1}^N \phi(\mathrm{Re}(C_{y_i} R h)) + \phi(\mathrm{Im}(C_{y_i} R h)), \quad \text{s.t. } \|h\| = 1,$$

where $R := (\frac{1}{\theta n N} \sum_{i=1}^N C_{y_i}^{\mathrm{H}} C_{y_i})^{-1/2} \in \mathbb{C}^{n \times n}$, and $(\cdot)^{\mathrm{H}}$ represents the Hermitian transpose. If one treats the real and imaginary parts of $h$ separately, then this optimization in $\mathbb{C}^n$ can be recast into $\mathbb{R}^{2n}$, and the gradient with respect to $\mathrm{Re}(h)$ and $\mathrm{Im}(h)$ can be used in first-order methods. This is related to Wirtinger gradient descent algorithms (see the discussion in [56]). The Riemannian gradient with respect to $h$ is $P_{(\mathbb{R}\cdot h)^\perp}(\frac{1}{N} \sum_{i=1}^N R^{\mathrm{H}} C_{y_i}^{\mathrm{H}} w_i(h))$, where $w_i(h)$ represents $w_i(h) = \nabla_\phi(\mathrm{Re}(C_{y_i} R h)) + \sqrt{-1}\nabla_\phi(\mathrm{Im}(C_{y_i} R h))$, and $P_{(\mathbb{R}\cdot h)^\perp}$ represents the projection onto the tangent space at $h$ in $S^{2n-1} \subset \mathbb{R}^{2n}$: $P_{(\mathbb{R}\cdot h)^\perp} z = z - \mathrm{Re}(h^{\mathrm{H}} z) \cdot h$. In the complex case, one can initialize the manifold gradient descent algorithm with a random $h^{(0)}$, for which $[\mathrm{Re}(h^{(0)})^\top, \mathrm{Im}(h^{(0)})^\top]^\top$ follows a uniform distribution on $S^{2n-1}$.

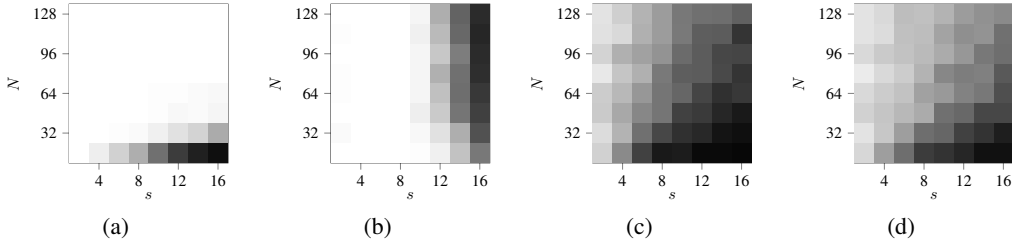

Figure 5: Empirical phase transition of $N$ versus $s$, given that $n = 128$. (a) Manifold gradient descent. (b) Truncated power iteration [31]. (c) Off-the-grid algebraic method [57]. (d) Off-the-grid optimization approach [58].

We compare manifold gradient descent (with random initialization) with three blind calibration algorithms that solve MSBD in the frequency domain: (i) truncated power iteration [31] (initialized with $f^{(0)} = e_1$ and $x_i^{(0)} = 0$); (ii) an off-the-grid algebraic method [57] (simplified from [55]); and (iii) an off-the-grid optimization approach [58].

We consider Gaussian random $\tilde{f} \sim CN(\mathbf{0}_{n \times 1}, I_n)$, and jointly $s$-sparse $\{x_i\}_{i=1}^N$, for which the support is chosen uniformly at random, and the nonzero entries of $\{x_i\}_{i=1}^N$ follow a complex Gaussian distribution $CN(0, 1)$. We fix $n = 128$, and run experiments for $N = 16, 32, 48, \cdots, 128$, and $s = 2, 4, 6, \ldots, 16$. We say that the recovery is successful is the accuracy (cosine of the angle between the true signal and the recovered signal) is greater than $0.7$.

By the phase transitions in Figure 5, manifold gradient descent and truncated power iteration are both successful when $N \geq 48$ and $s \leq 8$. However, although truncated power iteration achieves higher success rates when both $N$ and $s$ are small, it fails for $s > 8$ even with a large $N$. In contrast, manifold gradient descent can recover channels with $s = 16$ when $N \geq 80$.

The off-the-grid methods are designed, hence provide better recovery than the first two algorithms, for the case that the unknown sparse signals do *not* reside on a discrete grid (i.e., "off the grid"). However, the off-the-grid methods rely on the properties of the covariance matrix $\frac{1}{N} \sum_{i=1}^N y_i y_i^{\mathrm{H}}$, and require a much larger $N$ than the first two algorithms to achieve high success rates when the sparse signals actually lie on a regular grid (see the phase transitions in Figure 5).

## 5.3 Super-Resolution Fluorescence Microscopy

Manifold gradient descent can be applied to deconvolution of time resolved fluorescence microscopy images. The goal is to recover sharp images $x_i$'s from observations $y_i$'s that are blurred by an unknown PSF $f$.

We use a publicly available microtubule dataset [28], which contains $N = 626$ images (Figure 6(a)). Since fluorophores are are turned on and off stochastically, the images $x_i$'s are random sparse samples of the $64 \times 64$ microtubule image (Figure 6(e)). The observations $y_i$'s (Figure 6(b), 6(f)) are synthesized by circular convolutions with the PSF in Figure 6(i). The recovered images (Figure 6(c), 6(g)) and kernel (Figure 6(j)) clearly demonstrate the effectiveness of our approach in this setting.

Blind deconvolution is less sensitive to instrument calibration error than non-blind deconvolution. If the PSF used in a non-blind deconvolution method fails to account for certain optic aberration, the resulting images may suffer from spurious artifacts. For example, if we use a miscalibrated PSF (Figure 6(k)) in non-blind image reconstruction using FISTA [59], then the recovered images (Figure 6(d), 6(h)) suffer from serious spurious artifacts.

## 6 Conclusion

In this paper, we study the geometric structure of multichannel sparse blind deconvolution over the unit sphere. Our theoretical analysis reveals that local minima of a sparsity promoting smooth objective function correspond to signed shifted version of the ground truth, and saddle points have strictly negative curvatures. Thanks to the favorable geometric properties of the objective, we can

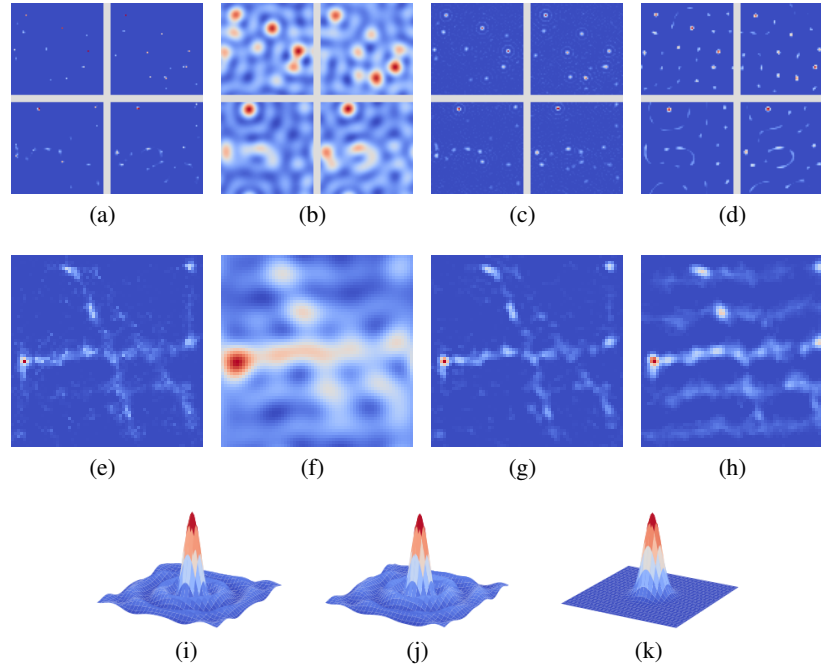

Figure 6: Super-resolution fluorescence microscopy experiment using manifold gradient descent. (a) True images. (b) Observed images. (c) Recovered images using blind deconvolution. (d) Recovered images using non-blind deconvolution and a miscalibrated PSF. (e)(f)(g)(h) are average images of (a)(b)(c)(d). (i) True PSF. (j) Recovered PSF using blind deconvolution. (k) Miscalibrated PSF used in non-blind deconvolution. All images in this figure are of the same size ($64 \times 64$).

simultaneously recover the unknown signal and unknown channels from convolutional measurements using manifold gradient descent with a random initialization. In practice, many convolutional measurement models are subsampled in the spatial domain (e.g., image super-resolution) or in the frequency domain (e.g., radio astronomy). Studying the effect of subsampling on the geometric structure of multichannel sparse blind deconvolution is an interesting problem for future work.

### Acknowledgments

This work was supported in part by the National Science Foundation (NSF) under Grant IIS 14-47879. The authors would like to thank Ju Sun for helpful discussions about this paper. The manuscript benefited from constructive comments by the anonymous reviewers.

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
