[Supplementary Material · geometry_neurips_2018_supplementary.pdf]

# Global Geometry of Multichannel Sparse Blind Deconvolution on the Sphere Supplementary Material

**Yanjun Li**
CSL and Department of ECE
University of Illinois
Urbana-Champaign
yli145@illinois.edu

**Yoram Bresler**
CSL and Department of ECE
University of Illinois
Urbana-Champaign
ybresler@illinois.edu

## 1 Proof of Theorem 3.1

Note that $R = (\frac{1}{\theta n N} \sum_{i=1}^{N} C_{y_i}^\top C_{y_i})^{-1/2}$ asymptotically converges to $(C_f^\top C_f)^{-1/2}$ as $N$ increases. Therefore, $L(h)$ can be approximated by

$$L'(h) = \frac{1}{N} \sum_{i=1}^{N} \phi(C_{y_i}(C_f^\top C_f)^{-1/2} h) = \frac{1}{N} \sum_{i=1}^{N} \phi(C_{x_i} C_f (C_f^\top C_f)^{-1/2} h).$$

Since $C_f(C_f^\top C_f)^{-1/2}$ is an orthogonal matrix, one can study the following objective function by rotating on the sphere $h' = C_f(C_f^\top C_f)^{-1/2} h$:

$$L''(h') = \frac{1}{N} \sum_{i=1}^{N} \phi(C_{x_i} h').$$

Our analysis consists of three parts: (1) geometric structure of $\mathbb{E}L''$, (2) deviation of $L''$ (or its rotated version $L'$) from its expectation $\mathbb{E}L''$, and (3) difference between $L$ and $L'$.

**Geometric structure of $\mathbb{E}L''$.** By the Bernoulli-Rademacher model (A1), the Riemannian gradient for $h \in S^{n-1}$ is computed as

$$\mathbb{E}\widehat{\nabla}_{L''}(h) = P_{h^\perp} \mathbb{E}\nabla_{L''}(h) = n\theta(1-3\theta)(\|h\|_4^4 \cdot h - h^{\odot 3}). \tag{1}$$

We use $x^{\odot k}$ to denote the entrywise $k$-th power of $x$. The Riemannian Hessian is

$$\mathbb{E}\widehat{H}_{L''}(h) = P_{h^\perp} \mathbb{E}H_{L''}(h) P_{h^\perp} - h^\top \mathbb{E}\nabla_{L''}(h) \cdot P_{h^\perp}$$
$$= n\theta(1-3\theta)\big[\|h\|_4^4 \cdot I + 2\|h\|_4^4 \cdot hh^\top - 3 \cdot \mathrm{diag}(h^{\odot 2})\big]. \tag{2}$$

Details of the derivation of (1) and (2) can be found in Appendix B.

At a stationary point of $\mathbb{E}L''(h)$ on $S^{n-1}$, the Riemannian gradient is zero. Since

$$\|\mathbb{E}\widehat{\nabla}_{L''}(h)\| = n\theta(1-3\theta)\sqrt{\|h\|_6^6 - \|h\|_4^8}$$
$$= n\theta(1-3\theta)\sqrt{\sum_{1 \le j < k \le n} h_{(j)}^2 h_{(k)}^2 (h_{(j)}^2 - h_{(k)}^2)^2}, \tag{3}$$

all nonzero entries of a stationary point $h_0$ have the same absolute value. Equivalently, $h_{0(j)} = \pm 1/\sqrt{r}$ if $j \in \Omega$ and $h_{0(j)} = 0$ if $j \notin \Omega$, for some $r \in [n]$ and $\Omega \subset [n]$ such that $|\Omega| = r$. Without

loss of generality (as justified below), we focus on stationary points that satisfy $h_{0(j)} = 1/\sqrt{r}$ if $j \in \{1, 2, \ldots, r\}$ and $h_{0(j)} = 0$ if $j \in \{r+1, \ldots, n\}$. The Riemannian Hessian at these stationary points is

$$\mathbb{E}\widehat{H}_{L''}(h_0) = \frac{n\theta(1-3\theta)}{r} \begin{bmatrix} \frac{2}{r}\mathbf{1}_{r\times r} - 2I_r & \mathbf{0}_{r\times(n-r)} \\ \mathbf{0}_{(n-r)\times r} & I_{n-r} \end{bmatrix}. \tag{4}$$

When $r = 1$, $h_0 = [1, 0, 0, \ldots, 0]^\top$, we have $\mathbb{E}\widehat{H}_{L''}(h_0) = n\theta(1-3\theta)P_{h_0^\perp}$. This Riemannian Hessian is positive definite on the tangent space,

$$\min_{\substack{z:\|z\|=1 \\ z\perp h_0}} z^\top \mathbb{E}\widehat{H}_{L''}(h_0)z = n\theta(1-3\theta) > 0. \tag{5}$$

Therefore, stationary points with one nonzero entry are local minima.

When $r > 1$, the Riemannian Hessian has at least one strictly negative eigenvalue:

$$\min_{\substack{z:\|z\|=1 \\ z\perp h_0}} z^\top \mathbb{E}\widehat{H}_{L''}(h_0)z = -\frac{2n\theta(1-3\theta)}{r} < 0. \tag{6}$$

Therefore, stationary points with more than one nonzero entry are strict saddle points, which, by definition, have at least one negative curvature direction on $S^{n-1}$. One such negative curvature direction satisfies $z_{(1)} = (r-1)/\sqrt{r(r-1)}$, $z_{(j)} = -1/\sqrt{r(r-1)}$ for $j \in \{2, 3, \ldots, r\}$, and $z_{(j)} = 0$ for $j \in \{r+1, \ldots, n\}$.

The Riemannian Hessian at other stationary points (different from the above stationary points by permutations and sign changes) can be computed similarly. By (2), a permutation and sign changes of the entries in $h_0$ has no effect on the bounds in (5) and (6), because the eigenvector $z$ that attains the minimum undergoes the same permutation and sign changes as $h_0$.

Next, in Lemma 1.2, we show that the properties of positive definiteness and negative curvature not only hold at the stationary points, but also hold in their neighborhoods defined as follows.

**Definition 1.1.** *We say that a point $h$ is in the $(\rho, r)$-neighborhood of a stationary point $h_0$ of $\mathbb{E}L''(h)$ with $r$ nonzero entries, if $\|h^{\odot 2} - h_0^{\odot 2}\|_\infty \leq \frac{\rho}{r}$. We define three sets:*

$\mathcal{H}_1'' := \{$*Points in the $(\rho, 1)$-neighborhoods of stationary points with 1 nonzero entry*$\}$,

$\mathcal{H}_2'' := \{$*Points in the $(\rho, r)$-neighborhoods of stationary points with $r > 1$ nonzero entries*$\}$,

$\mathcal{H}_3'' := S^{n-1} \backslash (\mathcal{H}_1'' \cup \mathcal{H}_2'')$.

*Clearly, $\mathcal{H}_1 \cap \mathcal{H}_2 = \emptyset$ for $\rho < 1/3$, hence $\mathcal{H}_1''$, $\mathcal{H}_2''$, and $\mathcal{H}_3''$ form a partition of $S^{n-1}$.*

**Lemma 1.2.** *Assume that positive constants $\theta < 1/3$, and $\rho < 10^{-3}$. Then*

○ *For $h \in \mathcal{H}_1''$,*

$$\min_{\substack{z:\|z\|=1 \\ z\perp h}} z^\top \mathbb{E}\widehat{H}_{L''}(h)z \geq n\theta(1-3\theta)(1-24\sqrt{\rho}) > 0. \tag{7}$$

○ *For $h \in \mathcal{H}_2''$,*

$$\min_{\substack{z:\|z\|=1 \\ z\perp h}} z^\top \mathbb{E}\widehat{H}_{L''}(h)z \leq -\frac{n\theta(1-3\theta)(2-24\sqrt{\rho})}{r} < 0. \tag{8}$$

○ *For $h \in \mathcal{H}_3''$,*

$$\|\mathbb{E}\widehat{\nabla}_{L''}(h)\| \geq \frac{\theta(1-3\theta)\rho^2}{n} > 0. \tag{9}$$

Lemma 1.2, and all other lemmas, are proved in the Appendix.

**Deviation of $L''$ from $\mathbb{E}L''$.** As the number $N$ of channels increases, the objective function $L''$ asymptotically converges to its expected value $\mathbb{E}L''$. Therefore, we can establish the geometric structure of $L''$ based on its similarity to $\mathbb{E}L''$. To this end, we give the following result.

**Lemma 1.3.** *Suppose that $\theta < 1/3$. There exist constants $c_1, c_1' > 0$ (depending only on $\theta$), such that: if $N > \frac{c_1 n^9}{\rho^4} \log \frac{n}{\rho}$, then with probability at least $1 - n^{-c_1'}$,*

$$\sup_{h \in S^{n-1}} \|\widehat{\nabla}_{L''}(h) - \mathbb{E}\widehat{\nabla}_{L''}(h)\| \leq \frac{\theta(1 - 3\theta)\rho^2}{4n},$$

$$\sup_{h \in S^{n-1}} \|\widehat{H}_{L''}(h) - \mathbb{E}\widehat{H}_{L''}(h)\| \leq \frac{\theta(1 - 3\theta)\rho^2}{n}.$$

By Lemma 1.3, the deviations from the corresponding expected values of the Riemannian gradient and Hessian due to a finite number of random $x_i$'s are small compared to the bounds in Lemma 1.2. Therefore, the Rimannian Hessian of $L''$ is still positive definite in the neighborhood of local minima, and has at least one strictly negative eigenvalue in the neighborhood of strict saddle points; and the Riemannian gradient of $L''$ is nonzero for all other points on the sphere. Since $L'$ and $L''$ differ only by an orthogonal matrix transformation of their argument, the geometric structure of $L'$ is identical to that of $L''$ up to a rotation on the sphere.

**Difference between $L$ and $L'$.** Recall that $L$ asymptotically converges to $L'$ as $N$ increases. The following result bounds the difference for a finite $N$.

**Lemma 1.4.** *Suppose that $\frac{1}{n} \leq \theta < \frac{1}{3}$. There exist constants $c_2, c_2' > 0$ (depending only on $\theta$), such that: if $N > \frac{c_2 \kappa^8 n^8}{\rho^4} \log n$, then with probability at least $1 - n^{-c_2'}$,*

$$\sup_{h \in S^{n-1}} \|\widehat{\nabla}_L(h) - \widehat{\nabla}_{L'}(h)\| \leq \frac{\theta(1 - 3\theta)\rho^2}{4n},$$

$$\sup_{h \in S^{n-1}} \|\widehat{H}_L(h) - \widehat{H}_{L'}(h)\| \leq \frac{\theta(1 - 3\theta)\rho^2}{n}.$$

We use $(C_f^\top C_f)^{1/2} C_f^{-1} \mathcal{H} = \{(C_f^\top C_f)^{1/2} C_f^{-1} h : h \in \mathcal{H}\}$ to denote the rotation of a set $\mathcal{H}$ by the orthogonal matrix $(C_f^\top C_f)^{1/2} C_f^{-1}$. Define the rotations of $\mathcal{H}_1''$, $\mathcal{H}_2''$, and $\mathcal{H}_3''$:

$$\mathcal{H}_1 := (C_f^\top C_f)^{1/2} C_f^{-1} \mathcal{H}_1'', \quad \mathcal{H}_2 := (C_f^\top C_f)^{1/2} C_f^{-1} \mathcal{H}_2'', \quad \mathcal{H}_3 := (C_f^\top C_f)^{1/2} C_f^{-1} \mathcal{H}_3''. \quad (10)$$

Combining Lemmas 1.2, 1.3, and 1.4, and the rotation relation between $L'$ and $L''$, we have:

- For $h \in \mathcal{H}_1$, the Riemannian Hessian is positive definite:

$$\min_{\substack{z:\|z\|=1 \\ z \perp h}} z^\top \widehat{H}_L(h) z \geq n\theta(1 - 3\theta)(1 - 24\sqrt{\rho} - \frac{2\rho^2}{n^2}) > 0.$$

- For $h \in \mathcal{H}_2$, the Riemannian Hessian has a strictly negative eigenvalue:

$$\min_{\substack{z:\|z\|=1 \\ z \perp h}} z^\top \widehat{H}_L(h) z \leq -\frac{n\theta(1 - 3\theta)(2 - 24\sqrt{\rho} - 2r\rho^2/n^2)}{r} < 0.$$

- For $h \in \mathcal{H}_3$, the Riemannian gradient is nonzero:

$$\|\widehat{\nabla}_L(h)\| \geq \frac{\theta(1 - 3\theta)\rho^2}{2n} > 0.$$

Clearly, all the local minima of $L(h)$ on $S^{n-1}$ belong to $\mathcal{H}_1$, and all the other stationary points are strict saddle points and belong to $\mathcal{H}_2$. The bounds in Theorem 3.1 on the Riemannian Hessian and the Riemannian gradient follows by setting

$$c(n, \theta, \rho) := \frac{\theta(1 - 3\theta)\rho^2}{2n}. \quad (11)$$

We complete the proof of Theorem 3.1 by giving the following result about $\mathcal{H}_1$.

**Lemma 1.5.** *If $h^* \in \mathcal{H}_1$, then for some $j \in [n]$,*

$$\|C_f R h^* \pm e_j\| \leq 2\kappa\sqrt{\rho}.$$

## 2 Proof of Theorem 4.1

We first establish that after $T$ steps, the iterate $h^{(T)} \in \mathcal{H}_1 \cup \mathcal{H}_2$, by applying [1, Theorem 4]. To this end, one needs to show that **(C1)** $L(h)$ has a finite lower bound, and that **(C2)** the function $\widehat{L}(z) := L(\frac{h+z}{\|h+z\|})$ (defined on $\{z : z \perp h\}$) is well approximated by its first-order Taylor expansion at $z = 0$. We verify conditions **(C1)** and **(C2)** in the following lemmas.

**Lemma 2.1.** *For all $h \in S^{n-1}$, $-4n^3 \leq L(h) \leq 0$, $\|\nabla_L(h)\| \leq 16n^3$, $\|H_L(h)\| \leq 48n^3$.*

**Lemma 2.2.** *Let $\widehat{L}(z) := L(\frac{h+z}{\|h+z\|})$. Then for all $z \perp h$,*

$$\left| \widehat{L}(z) - \widehat{L}(0) - \langle z, \nabla_{\widehat{L}}(0) \rangle \right| \leq 64n^3 \|z\|^2.$$

By [1, Theorem 4] and Lemmas 2.1 and 2.2, manifold gradient decent with a fixed step size $\gamma = 1/(2 \times 64n^3)$ achieves $\|\widehat{\nabla}_L(h^{(t)})\| < \tau$ after $t = 2[L(h^{(0)}) - \min_{h \in S^{n-1}} L(h)]/(\gamma\tau^2)$ iterations. Setting $\tau = \theta(1 - 3\theta)\rho^2/(2n)$ and $T = 4096n^8/[\theta^2(1 - 3\theta)^2\rho^4]$, it follows that

$$\|\widehat{\nabla}_L(h^{(t)})\| < \frac{\theta(1 - 3\theta)\rho^2}{2n} = c(n, \theta, \rho)$$

after $t \geq T$ iterations. By Theorem 3.1, we have $\{h^{(t)}\}_{t \geq T} \subset \mathcal{H}_1 \cup \mathcal{H}_2$. Since the distance between every pair of points $h_1 \in \mathcal{H}_1$ and $h_2 \in \mathcal{H}_2$ satisfies $\|h_1 - h_2\| \gg \gamma\|\widehat{\nabla}_L(h^{(t)})\|$, the iterates $\{h^{(t)}\}_{t \geq T}$ all belong to $\mathcal{H}_1$ or all belong to $\mathcal{H}_2$, and cannot jump from one set to the other.

Next, we show that if the initialization $h^{(0)}$ follows a random distribution on $S^{n-1}$, then $h^{(T)} \in \mathcal{H}_1$ almost surely, by applying [2, Theorem 2]. To this end, we verify that **(C3)** the strict saddle points are unstable fixed points of manifold gradient descent, and that **(C4)** the differential of $\mathcal{A}(\cdot)$ (defined in (2) in the paper) is invertible.

Let $h' = h - \gamma\widehat{\nabla}_L(h)$. The differential $D\mathcal{A}(h)$, as defined in [2, Definition 4], is

$$D\mathcal{A}(h) = P_{h'^\perp} P_{h^\perp} [I - \gamma\widehat{H}_L(h)] P_{h^\perp}. \tag{12}$$

At strict saddle points $\widehat{\nabla}_L(h) = 0$ and $h' = h$. Because, as we have shown, $\widehat{H}_L(h)$ has a strictly negative eigenvalue, it follows from [2, Proposition 8] that $D\mathcal{A}(h)$ has at least one eigenvalue larger than 1. Therefore, strict saddle points are unstable fixed points of manifold gradient descent (see [2, Definition 5]), i.e., **(C3)** is satisfied.

We verify **(C4)** in the following lemma.

**Lemma 2.3.** *For step size $\gamma = \frac{1}{128n^3}$, and all $h \in S^{n-1}$, we have $\det(D\mathcal{A}(h)) \neq 0$.*

Since conditions **(C3)** and **(C4)** are satisfied, by [2, Theorem 2], the set of initial points that converge to strict saddle points have measure 0. Therefore, a random $h^{(0)}$ uniformly distributed on $S^{n-1}$ converges to a local minimum almost surely. Hence $\{h^{(t)}\}_{t \geq T} \subset \mathcal{H}_1$. By Lemma 1.5,

$$\|C_f Rh^{(T)} \pm e_j\| \leq 2\kappa\sqrt{\rho},$$

for some $j \in [n]$.

## 3 Other Numerical Experiments

### 3.1 2D Deconvolution

Next, we run a numerical experiment with blind image deconvolution. Suppose the circular convolutions $\{y_i\}_{i=1}^N$ (Figure 1(c)) of an unknown image $f$ (Figure 1(a)) and unknown sparse channels $\{x_i\}_{i=1}^N$ (Figure 1(b)) are observed. The recovered image $\hat{f}$ (Figure 1(d)) is computed as follows:

$$\hat{f} = \mathcal{F}^{-1}\big[\mathcal{F}(Rh^{(T)})^{\odot-1}\big],$$

where $\mathcal{F}$ denotes the 2D DFT, and $h^{(T)}$ is the output of manifold gradient descent, with a random initialization $h^{(0)}$ that is uniformly distributed on the sphere.

Figure 1 shows that, although the sparse channels are completely unknown and the convolutional observations have corrupted the image beyond recognition, manifold gradient descent is capable of recovering a shifted version of the (negative) image, starting from a random point on the sphere (see the image recovered using a random initialization in Figure 1(d), and then corrected with the true sign and shift in Figure 1(e)). In this example, all images and channels are of size $64 \times 64$, the number of channels is $N = 256$, and the sparsity level is $\theta = 0.01$. We run $T = 100$ iterations of manifold gradient descent with a fixed step size $\gamma = 0.05$. The accuracy $\frac{\|C_f R h^{(t)}\|_\infty}{\|C_f R h^{(t)}\|}$ as a function of iteration number $t$ is shown in Figure 1(f), and exhibits a sharp transition at a modest number ($\approx 80$) of iterations.

Figure 1: Multichannel blind image deconvolution. (a) True image. (b) Sparse channels. (c) Observations. (d) Recovered image using manifold gradient descent. (e) Recovered image with sign and shift correction. (f) The accuracy as a function of iteration number. All images and channels in this figure are of the same size ($64 \times 64$).

## 3.2 MSBD with a Linear Convolution Model

In this section, we empirically study MSBD with a linear convolution model. Suppose the observations $y_i = x'_i * f' \in \mathbb{R}^n$ ($i = 1, 2, \ldots, N$) are linear convolutions of $s$-sparse channels $x'_i \in \mathbb{R}^m$ and a signal $f' \in \mathbb{R}^{n-m+1}$. Let $x_i \in \mathbb{R}^n$ and $f \in \mathbb{R}^n$ denote the zero-padded versions of $x'_i$ and $f$. Then

$$y_i = x'_i * f' = x_i \circledast f.$$

In this section, we show that one can solve for $f$ and $x_i$ using the optimization formulation (P1) and the manifold gradient descent algorithm, without knowledge of the length $m$ of the channels.

We compare our approach to the subspace method based on cross convolution [3], which solves for the concatenation of the channels as a null vector of a structured matrix. For fairness, we also compare to an alternative method that takes advantage of the sparsity of the channels, and finds a sparse null vector of the same structured matrix as in [3], using truncated power iteration [4, 5].[1]

In our experiments, we synthesize $f'$ using a random Gaussian vector following $N(\mathbf{0}_{(n-m+1)\times 1}, I_{n-m+1})$. We synthesize $s$-sparse channels $x_i$ such that the support is chosen uniformly at random, and the nonzero entries are independent following $N(0, 1)$. We denote the zero-padded versions of the true signal and the recovered signal by $f$ and $\hat{f}$, and declare success if the accuracy (the cosine of the angle between the true signal and the recovered signal) is greater than 0.7. We study the empirical success rates of our method and the competing methods in three experiments:

- $N$ versus $s$, given that $n = 128$ and $m = 64$.
- $N$ versus $m$, given that $n = 128$ and $s = 4$.
- $N$ versus $n$, given that $m = 64$ and $s = 4$.

The phase transitions in Figure 2 show that our manifold gradient descent method consistently has higher success rates than the competing methods based on cross convolution. The subspace method and the truncated power iteration method are only successful when $m$ is small compared to $n$, while our method is successful for a large range of $m$ and $n$. The sparsity prior exploited by truncated power iteration improves the success rate over the subspace method, but only when the sparsity level $s$ is small compared to $m$. In comparison, our method, given a sufficiently large number $N$ of channels, can recover channels with a much larger $s$.

Figure 2: Empirical phase transition of MSBD with a linear convolution model. The first row shows the phase transitions of $N$ versus $s$. The second row shows the phase transitions of $N$ versus $m$. The third row shows the phase transitions of $N$ versus $n$. The first column shows the results for manifold gradient descent. The second column shows the results for the subspace method [3]. The third column shows are the results for truncated power iteration.

# A  Proof of Corollary 4.2

Since $\|C_f R\hat{h} \pm e_j\| \le 2\kappa\sqrt{\rho}$ for some $j \in [n]$, by the Cauchy-Schwarz inequality

$$\|\mathcal{F}(f) \odot \mathcal{F}(R\hat{h}) - \mathcal{F}(\mp e_j)\|_\infty \le \sqrt{n}\|C_f R\hat{h} \pm e_j\| \le 2\kappa\sqrt{\rho n}, \tag{13}$$

where $x \odot y$ denotes the entrywise product between $x$ and $y$. Equivalently, the circular convolution operators satisfy
$$\|C_f C_{R\hat{h}} - C_{\mp e_j}\| \le 2\kappa\sqrt{\rho n}.$$
It follows that
$$\begin{aligned}
\|\hat{x}_i \pm \mathcal{S}_j(x_i)\| &= \|C_{y_i} R\hat{h} \pm \mathcal{S}_j(x_i)\| \\
&= \|C_f C_{R\hat{h}} x_i - C_{\mp e_j} x_i\| \le \|C_f C_{R\hat{h}} - C_{\mp e_j}\| \cdot \|x_i\| \\
&\le 2\kappa\sqrt{\rho n} \cdot \|x_i\|.
\end{aligned}$$

The smallest singular value $\sigma_n(C_{R\hat{h}})$ of $C_{R\hat{h}}$ equals the smallest magnitude of the DFT $\mathcal{F}(R\hat{h})$. The largest singular value of $C_f$ equals the largest DFT magnitude $\|\mathcal{F}(f)\|_\infty \le \sqrt{n}\|f\|$. Therefore, it follows from (13) that
$$\sigma_n(C_{R\hat{h}}) \ge \frac{1 - 2\kappa\sqrt{\rho n}}{\|\mathcal{F}(f)\|_\infty} \ge \frac{1 - 2\kappa\sqrt{\rho n}}{\sqrt{n}\|f\|}.$$
Combining the above bound with the following
$$\|C_{R\hat{h}}(f \pm \mathcal{S}_j(\hat{f}))\| = \|C_f R\hat{h} \pm e_j\| \le 2\kappa\sqrt{\rho},$$
we have
$$\begin{aligned}
\|\hat{f} \pm \mathcal{S}_{-j}(f)\| &= \|f \pm \mathcal{S}_j(\hat{f})\| \\
&= \frac{\|C_{R\hat{h}}(f \pm \mathcal{S}_j(\hat{f}))\|}{\sigma_n(C_{R\hat{h}})} \le 2\kappa\sqrt{\rho} \times \frac{\sqrt{n}\|f\|}{1 - 2\kappa\sqrt{\rho n}} \\
&= \frac{2\kappa\sqrt{\rho n}}{1 - 2\kappa\sqrt{\rho n}} \cdot \|f\|,
\end{aligned}$$

# B  Derivation of (1) and (2)

Recall that
$$\nabla_{L''}(h) = \frac{1}{N}\sum_{i=1}^N \nabla_i'',$$
$$H_{L''}(h) = \frac{1}{N}\sum_{i=1}^N H_i'',$$
where $\nabla_i'' := C_{x_i}^\top \nabla_\phi(C_{x_i}h)$, and $H_i'' = C_{x_i}^\top H_\phi(C_{x_i}h)C_{x_i}$.

For the Bernoulli-Rademacher model in (A1), we have
$$\begin{aligned}
\mathbb{E}\nabla_{i(j)}'' &= -\mathbb{E}\sum_{s=1}^n x_{i(1+s-j)}\left(\sum_{t=1}^n x_{i(1+s-t)}h_{(t)}\right)^3 \\
&= -n\left(\theta h_{(j)}^3 + 3\theta^2 h_{(j)}\sum_{\ell \ne j} h_{(\ell)}^2\right) \\
&= -n\theta(1 - 3\theta)h_{(j)}^3 - 3n\theta^2 h_{(j)},
\end{aligned}$$
where the last line uses the fact that $\sum_{j=1}^n h_{(j)}^2 = \|h\| = 1$. Therefore, the gradient and the Riemannian gradient are
$$\mathbb{E}\nabla_{L''}(h) = -n\theta(1 - 3\theta)h^{\odot 3} - 3n\theta^2 h,$$
$$\mathbb{E}\widehat{\nabla}_{L''}(h) = P_{h^\perp}\mathbb{E}\nabla_{L''}(h) = n\theta(1 - 3\theta)(\|h\|_4^4 \cdot h - h^{\odot 3}).$$

Similarly, we have
$$\begin{aligned}
\mathbb{E}H_{i(jk)}'' &= -3\mathbb{E}\sum_{s=1}^n x_{i(1+s-j)}x_{i(1+s-k)}\left(\sum_{t=1}^n x_{i(1+s-t)}h_{(t)}\right)^2 \\
&= -3n \times \begin{cases} \theta h_{(j)}^2 + \theta^2 \sum_{\ell \ne j} h_{(\ell)}^2 & \text{if } j = k \\ 2\theta^2 h_{(j)}h_{(k)} & \text{if } j \ne k \end{cases} \\
&= -3n\left[\theta^2 \delta_{jk} + \theta(1 - 3\theta)h_{(j)}^2 \delta_{jk} + 2\theta^2 h_{(j)}h_{(k)}\right].
\end{aligned}$$

The Hessian and the Riemannian Hessian are

$$\mathbb{E}H_{L''}(h) = -3n\big[\theta^2 I + \theta(1 - 3\theta)\mathrm{diag}(h^{\odot 2}) + 2\theta^2 hh^\top\big],$$

$$\mathbb{E}\widehat{H}_{L''}(h) = P_{h\perp}\mathbb{E}H_{L''}(h)P_{h\perp} - h^\top\mathbb{E}\nabla_{L''}(h)\cdot P_{h\perp}$$

$$= n\theta(1 - 3\theta)\big[\|h\|_4^4 \cdot I + 2\|h\|_4^4 \cdot hh^\top - 3\cdot\mathrm{diag}(h^{\odot 2})\big].$$

## C   Proofs of Lemmas 1.2 – 1.5

*Proof of Lemma 1.2.* We first investigate the Riemannian Hessian at points in $\mathcal{H}_1''$ and $\mathcal{H}_2''$. Without loss of generality, we consider points close to the representative stationary point $h_0 = [1/\sqrt{r}, \ldots, 1/\sqrt{r}, 0, \ldots, 0]$. We have

$$|h_{(j)}^2 - 1/r| \le \rho/r, \qquad \forall j \in \{1, 2, \ldots, r\},$$

$$h_{(j)}^2 \le \rho/r, \qquad \forall j \in \{r + 1, \ldots, n\},$$

$$\sum_{j=r+1}^{n} h_{(j)}^2 = 1 - \sum_{j=1}^{r} h_{(j)}^2 \le \rho.$$

Therefore,

$$\|h - h_0\| \le \sqrt{r \times \big(\frac{1 - \sqrt{1 - \rho}}{\sqrt{r}}\big)^2 + \rho} \le \sqrt{2\rho}, \tag{14}$$

$$\left\|\mathrm{diag}(h^{\odot 2}) - \frac{1}{r}\begin{bmatrix} I_r & \\ & \mathbf{0}_{(n-r)\times(n-r)} \end{bmatrix}\right\| \le \frac{\rho}{r}, \tag{15}$$

and

$$\left\|hh^\top - \frac{1}{r}\begin{bmatrix} \mathbf{1}_{r\times r} & \\ & \mathbf{0}_{(n-r)\times(n-r)} \end{bmatrix}\right\| \le 2\|h - h_0\| \le 2\sqrt{2\rho}. \tag{16}$$

We also bound $\|h\|_4^4$ as follows:

$$\|h\|_4^4 \le r \times \frac{(1 + \rho)^2}{r^2} + \min\Big\{(n - r) \times \frac{\rho^2}{r^2}, \frac{\rho^2}{n - r}\Big\} \le \frac{1 + 2\rho + 2\rho^2}{r},$$

$$\|h\|_4^4 \ge r \times \frac{(1 - \rho)^2}{r^2} \ge \frac{1 - 2\rho + \rho^2}{r}.$$

Since $\rho < 10^{-3} < 1/2$,

$$\left|\|h\|_4^4 - \frac{1}{r}\right| \le \frac{3\rho}{r}. \tag{17}$$

Next we obtain bounds on the Riemannian curvature of $\mathbb{E}L''$ at points $h \in \mathcal{H}_1''$ or $h \in \mathcal{H}_2''$ by bounding its deviation from the Riemannian curvature at a corresponding stationary point $h_0$. By (15), (16), (17), and the expressions in (2), (4):

$$\|\mathbb{E}\widehat{H}_{L''}(h) - \mathbb{E}\widehat{H}_{L''}(h_0)\|$$

$$\le n\theta(1 - 3\theta)\Big[\frac{3\rho}{r} + 2 \times \frac{3\rho + 2\sqrt{2\rho}}{r} + 3 \times \frac{\rho}{r}\Big]$$

$$= \frac{n\theta(1 - 3\theta)}{r}(12\rho + 4\sqrt{2\rho}). \tag{18}$$

It follows that

$$
\begin{aligned}
& \left| \min_{\substack{z:\|z\|=1 \\ z \perp h}} z^\top \mathbb{E}\widehat{H}_{L''}(h)z - \min_{\substack{z:\|z\|=1 \\ z \perp h_0}} z^\top \mathbb{E}\widehat{H}_{L''}(h_0)z \right| \\
& \leq \left| \min_{\substack{z:\|z\|=1 \\ z \perp h}} z^\top \mathbb{E}\widehat{H}_{L''}(h)z - \min_{\substack{z:\|z\|=1 \\ z \perp h}} z^\top \mathbb{E}\widehat{H}_{L''}(h_0)z \right| \\
& \quad + \left| \min_{\substack{z:\|z\|=1 \\ z \perp h}} z^\top \mathbb{E}\widehat{H}_{L''}(h_0)z - \min_{\substack{z:\|z\|=1 \\ z \perp h_0}} z^\top \mathbb{E}\widehat{H}_{L''}(h_0)z \right| \\
& \leq \|V^\top \mathbb{E}\widehat{H}_{L''}(h)V - V^\top \mathbb{E}\widehat{H}_{L''}(h_0)V\| + \|V^\top \mathbb{E}\widehat{H}_{L''}(h_0)V - V_0^\top \mathbb{E}\widehat{H}_{L''}(h_0)V_0\| \\
& \leq \|\mathbb{E}\widehat{H}_{L''}(h) - \mathbb{E}\widehat{H}_{L''}(h_0)\| + 2\|\mathbb{E}\widehat{H}_{L''}(h_0)\| \cdot \|V - V_0\| \\
& \leq \frac{n\theta(1-3\theta)}{r}(12\rho + 4\sqrt{2\rho}) + 2 \times \frac{2n\theta(1-3\theta)}{r} \times \sqrt{2\rho} \\
& = \frac{n\theta(1-3\theta)}{r}(12\rho + 8\sqrt{2\rho}) \\
& \leq \frac{n\theta(1-3\theta)(24\sqrt{\rho})}{r},
\end{aligned}
\tag{19}
$$

where $V, V_0 \in \mathbb{R}^{n \times (n-1)}$ satisfy: (I) the columns of $V$ (resp. $V_0$) form an orthonormal basis for the tangent space at $h$ (resp. $h_0$); (II) $\|V - V_0\| \leq \sqrt{2\rho}$. We construct $V$ and $V_0$ as follows, for the non-trivial case where $h \neq h_0$. Suppose the columns of $V_\cap \in \mathbb{R}^{n \times (n-2)}$ form an orthonormal basis for the intersection of the tangent spaces at $h$ and at $h_0$. Let $c := \langle h, h_0 \rangle < 1$, and let $h' := \frac{1}{\sqrt{1-c^2}}(h_0 - ch)$ and $h_0' := \frac{1}{\sqrt{1-c^2}}(ch_0 - h)$. It is easy to verify that $V := [V_\cap, h']$ and $V_0 := [V_\cap, h_0']$ satisfy (I). To verify (II), we have $\|V - V_0\| = \|h' - h_0'\| = \frac{1-c}{\sqrt{1-c^2}}\|h + h_0\| = \|h - h_0\| \leq \sqrt{2\rho}$.

Positive definiteness (7) follows from (5) and (19). Negative curvature (8) follows from (6) and (19).

Next, we prove contrapositive of (9), i.e., suppose $\|\mathbb{E}\widehat{\nabla}_{L''}(h)\| < \theta(1-3\theta)\rho^2/n$ for some $h \in S^{n-1}$, then we show $h \in \mathcal{H}_1'' \cup \mathcal{H}_2''$. First, it follows from $\|\mathbb{E}\widehat{\nabla}_{L''}(h)\| < \theta(1-3\theta)\rho^2/n$, and the expression in (3), that for all $j, k \in [n]$,

$$
h_{(j)}^2 h_{(k)}^2 (h_{(j)}^2 - h_{(k)}^2)^2 < \frac{\rho^4}{n^4}.
$$

As a result, $|h_{(j)}^2 - h_{(k)}^2| < \rho/n$ if $h_{(j)}^2 \geq \rho/n$ and $h_{(k)}^2 \geq \rho/n$.

Let $\Omega := \{j : h_{(j)}^2 \geq \rho/n\} \subset [n]$, and $r := |\Omega|$. Then

$$
h_{(j)}^2 < \rho/n \leq \rho/r, \qquad \forall j \notin \Omega,
\tag{20}
$$

and

$$
1 - (n-r) \cdot \frac{\rho}{n} < \sum_{j \in \Omega} h_{(j)}^2 \leq 1.
\tag{21}
$$

In addition, $|h_{(j)}^2 - h_{(k)}^2| < \rho/n$ for $j, k \in \Omega$. Therefore, for $k \in \Omega$, $h_{(k)}^2$ is close to the average $\frac{1}{r}\sum_{j \in \Omega} h_{(j)}^2$:

$$
\left| h_{(k)}^2 - \frac{1}{r}\sum_{j \in \Omega} h_{(j)}^2 \right| < \rho/n, \qquad \forall k \in \Omega
\tag{22}
$$

By (21) and (22), for $k \in \Omega$:

$$
h_{(k)}^2 \leq \frac{1}{r} + \frac{\rho}{n} \leq \frac{1+\rho}{r},
$$

$$
h_{(k)}^2 \geq \frac{1 - (n-r) \cdot \frac{\rho}{n}}{r} - \frac{\rho}{n} = \frac{1-\rho}{r}.
$$

Therefore,

$$\left| h_{(k)}^2 - \frac{1}{r} \right| \le \frac{\rho}{r} \qquad \forall k \in \Omega, \tag{23}$$

It follows from (20) and (23) that $h$ is in the $(\rho, r)$-neighborhood of a stationary point $h_0$, where $h_{0(j)} = 1/\sqrt{r}$ if $j \in \Omega$ and $h_{0(j)} = 0$ if $j \notin \Omega$. Clearly, such an $h$ belongs to $\mathcal{H}_1'' \cup \mathcal{H}_2''$. By contraposition, any point $h \in \mathcal{H}_3'' = S^{n-1} \backslash (\mathcal{H}_1'' \cup \mathcal{H}_2'')$ satisfies (9). $\qquad \square$

*Proof of Lemma 1.3.* For any given $h \in S^{n-1}$ one can bound the deviation of the gradient (or Hessian) from its mean using matrix Bernstein inequality [6]. Let $S_\epsilon$ be an $\epsilon$-net of $S^{n-1}$. Then $|S_\epsilon| \le (3/\epsilon)^n$ [7, Lemma 9.5]. We can then bound the deviation over $S^{n-1}$ by a union bound over $S_\epsilon$.

Define $\nabla_i'' := C_{x_i}^\top \nabla_\phi (C_{x_i} h)$, and $H_i'' = C_{x_i}^\top H_\phi (C_{x_i} h) C_{x_i}$. For the Bernoulli-Rademacher model in (A1), we have $|x_{i(j)}| \le 1$. Therefore,

$$\left| \nabla_{i(j)}'' \right| = \left| \sum_{s=1}^n x_{i(1+s-j)} \left( \sum_{t=1}^n x_{i(1+s-t)} h_{(t)} \right)^3 \right|$$

$$\le n \left( \sum_{t=1}^n |h_{(t)}| \right)^3$$

$$\le n^2 \sqrt{n},$$

$$\left| H_{i(jk)}'' \right| = \left| 3 \sum_{s=1}^n x_{i(1+s-j)} x_{i(1+s-k)} \left( \sum_{t=1}^n x_{i(1+s-t)} h_{(t)} \right)^2 \right|$$

$$\le 3n \left( \sum_{t=1}^n |h_{(t)}| \right)^2$$

$$\le 3n^2.$$

It follows that $\|\nabla_i''\| \le n^3$, and $\|H_i''\| \le \|H_i''\|_{\mathrm{F}} \le 3n^3$.

Our goal is to bound the following average of independent random terms with zero mean:

$$\nabla_{L''}(h) - \mathbb{E}\nabla_{L''}(h) = \frac{1}{N} \sum_{i=1}^N (\nabla_i'' - \mathbb{E}\nabla_i'').$$

$$H_{L''}(h) - \mathbb{E}H_{L''}(h) = \frac{1}{N} \sum_{i=1}^N (H_i'' - \mathbb{E}H_i'').$$

Since $\|\nabla_i''\| \le n^3$, we have

$$\|\nabla_i'' - \mathbb{E}\nabla_i''\| \le 2n^3,$$

$$\sum_{i=1}^N \mathbb{E}\|\nabla_i'' - \mathbb{E}\nabla_i''\|^2 \le N(\mathbb{E}\|\nabla_i''\|^2 - \|\mathbb{E}\nabla_i''\|^2) \le 2Nn^6,$$

$$\left\| \sum_{i=1}^N \mathbb{E}(\nabla_i'' - \mathbb{E}\nabla_i'')(\nabla_i'' - \mathbb{E}\nabla_i'')^\top \right\| \le N(\mathbb{E}\|\nabla_i''\|^2 + \|\mathbb{E}\nabla_i''\|^2) \le 2Nn^6.$$

By the rectangular version of the matrix Bernstein inequality [6, Theorem 1.6], and a union bound over $S_\epsilon$,

$$\mathbb{P}\left[ \sup_{h \in S_\epsilon} \|\nabla_{L''}(h) - \mathbb{E}\nabla_{L''}(h)\| \le \tau \right]$$

$$\ge 1 - \left( \frac{3}{\epsilon} \right)^n (n+1) \exp\left( \frac{-N^2\tau^2/2}{2Nn^6 + 2n^3 N\tau/3} \right) \tag{24}$$

Similarly, since $\|H_i''\| \le 3n^3$, we have
$$\|H_i'' - \mathbb{E}H_i''\| \le 6n^3,$$
$$\left\|\sum_{i=1}^N \mathbb{E}(H_i'' - \mathbb{E}H_i'')^2\right\| \le N\|\mathbb{E}H_i''^2 - (\mathbb{E}H_i'')^2\| \le 2N(3n^3)^2 = 18Nn^6.$$

By the symmetric version of the matrix Bernstein inequality [6, Theorem 1.4], and a union bound over $S_\epsilon$,
$$\mathbb{P}\left[\sup_{h \in S_\epsilon} \|H_{L''}(h) - \mathbb{E}H_{L''}(h)\| \le \tau\right]$$
$$\ge 1 - \left(\frac{3}{\epsilon}\right)^n (2n) \exp\left(\frac{-N^2\tau^2/2}{18Nn^6 + 6n^3N\tau/3}\right) \tag{25}$$

Choose $\tau = \frac{\theta(1-3\theta)\rho^2}{8n}$, and $\epsilon = \frac{\tau}{6n^3} = \frac{\theta(1-3\theta)\rho^2}{48n^4}$. By (24) and (25), there exist constants $c_1$, $c_1' > 0$ (depending only on $\theta$), such that: if $N > \frac{c_1 n^9}{\rho^4} \log \frac{n}{\rho}$, then with probability at least $1 - n^{-c_1'}$,
$$\sup_{h \in S_\epsilon} \|\nabla_{L''}(h) - \mathbb{E}\nabla_{L''}(h)\| \le \tau = \frac{\theta(1-3\theta)\rho^2}{8n},$$
$$\sup_{h \in S_\epsilon} \|H_{L''}(h) - \mathbb{E}H_{L''}(h)\| \le \tau = \frac{\theta(1-3\theta)\rho^2}{8n}.$$

To finish the proof, we extrapolate the concentration bounds over $S_\epsilon$ to all points in $S^{n-1}$. For any $h \in S^{n-1}$, there exists $h' \in S_\epsilon$ such that $\|h - h'\| \le \epsilon$. Furthermore, thanks to the Lipschitz continuity of the gradient and the Hessian,
$$\|\nabla_i''(h) - \nabla_i''(h')\|$$
$$\le \|C_{x_i}\| \cdot \sqrt{n}(3\|x_i\|^2) \cdot \|x_i\|\|h - h'\|$$
$$\le 3n^3\epsilon,$$

$$\|H_i''(h) - H_i''(h')\|$$
$$\le \|C_{x_i}\|^2 \cdot (6\|x_i\|) \cdot \|x_i\|\|h - h'\|$$
$$\le 6n^3\epsilon,$$

where $3\|x_i\|^2$ and $6\|x_i\|$ are the Lipschitz constants of $(\cdot)^3$ and $3(\cdot)^2$ on the interval $[-\|x_i\|, \|x_i\|]$. We also use the fact that $|x_{i(j)}| < 1$, hence $\|x_i\| \le \sqrt{n}$ and $\|C_{x_i}\| \le n$. As a consequence,
$$\sup_{h \in S^{n-1}} \|\nabla_{L''}(h) - \mathbb{E}\nabla_{L''}(h)\|$$
$$\le \sup_{h \in S_\epsilon} \|\nabla_{L''}(h) - \mathbb{E}\nabla_{L''}(h)\| + 2\max_{i \in [n]} \sup_{\|h-h'\| \le \epsilon} \|\nabla_i''(h) - \nabla_i''(h')\|$$
$$\le \tau + 6n^3\epsilon = 2\tau = \frac{\theta(1-3\theta)\rho^2}{4n},$$
$$\sup_{h \in S^{n-1}} \|\widehat{\nabla}_{L''}(h) - \mathbb{E}\widehat{\nabla}_{L''}(h)\|$$
$$\le \sup_{h \in S^{n-1}} \|\nabla_{L''}(h) - \mathbb{E}\nabla_{L''}(h)\|$$
$$\le \frac{\theta(1-3\theta)\rho^2}{4n}.$$

Similarly,
$$\sup_{h \in S^{n-1}} \|H_{L''}(h) - \mathbb{E}H_{L''}(h)\|$$
$$\le \sup_{h \in S_\epsilon} \|H_{L''}(h) - \mathbb{E}H_{L''}(h)\| + 2\max_{i \in [n]} \sup_{\|h-h'\| \le \epsilon} \|H_i''(h) - H_i''(h')\|$$
$$\le \tau + 12n^3\epsilon = 3\tau = \frac{3\theta(1-3\theta)\rho^2}{8n},$$

$$\sup_{h \in S^{n-1}} \|\widehat{H}_{L''}(h) - \mathbb{E}\widehat{H}_{L''}(h)\|$$

$$\leq \sup_{h \in S^{n-1}} \|H_{L''}(h) - \mathbb{E}H_{L''}(h)\| + \sup_{h \in S^{n-1}} \|\nabla_{L''}(h) - \mathbb{E}\nabla_{L''}(h)\|$$

$$\leq \frac{\theta(1 - 3\theta)\rho^2}{n}.$$

<div align="right">□</div>

*Proof of Lemma 1.4.* We have $\mathbb{E}\frac{1}{\theta n N} \sum_{i=1}^{N} C_{x_i}^\top C_{x_i} = I$. We first bound $\|\frac{1}{\theta n N} \sum_{i=1}^{N} C_{x_i}^\top C_{x_i} - I\|$ using the matrix Bernstein inequality. To this end, we bound the spectral norm of $\mathbb{E}(C_{x_i}^\top C_{x_i})^2$, the singular values of which can be computed using the DFT of $x_i$. The singular value corresponding to the $t$-th frequency satisfies

$$\mathbb{E}\Big[\Big(\sum_{k=1}^{N} e^{-\sqrt{-1}(k-1)t/n} x_{i(k)}\Big)\Big(\sum_{k=1}^{N} e^{\sqrt{-1}(k-1)t/n} x_{i(k)}\Big)\Big]^2$$

$$= \mathbb{E}\Big(\sum_{k=1}^{N} x_{i(k)}^2 + \sum_{1 \leq k < j \leq n} 2\cos((j-k)t/n)x_{i(j)}x_{i(k)}\Big)^2$$

$$\leq n\theta + \frac{n(n-1)}{2} \times 4\theta^2 + n(n-1)\theta^2$$

$$= n\theta + 3n(n-1)\theta^2.$$

Therefore,

$$\Big\|\sum_{i=1}^{N} \mathbb{E}\Big(\frac{1}{\theta n} C_{x_i}^\top C_{x_i} - I\Big)^2\Big\|$$

$$= N\Big\|\frac{1}{\theta^2 n^2}\mathbb{E}(C_{x_i}^\top C_{x_i})^2 - I\Big\|$$

$$\leq \frac{N}{\theta^2 n^2}\|\mathbb{E}(C_{x_i}^\top C_{x_i})^2\| + N$$

$$\leq \frac{N}{\theta^2 n^2}(n\theta + 3n(n-1)\theta^2) + N$$

$$\leq \frac{N}{\theta n} + 3N + N$$

$$\leq 5N.$$

We also have

$$\|\frac{1}{\theta n} C_{x_i}^\top C_{x_i} - I\| \leq \frac{1}{\theta n}\|C_{x_i}\|^2 + 1 \leq \frac{n^2}{\theta n} + 1 \leq n^2 + 1.$$

By the matrix Bernstein inequality [6, Theorem 1.4],

$$\mathbb{P}\Big[\Big\|\frac{1}{\theta n N} \sum_{i=1}^{N} C_{x_i}^\top C_{x_i} - I\Big\| \leq \tau\Big] \geq 1 - 2n\exp\Big(\frac{-N^2\tau^2/2}{5N + (n^2+1)N\tau/3}\Big).$$

Set $\tau = \frac{\theta(1-3\theta)\rho^2}{200n^4\kappa^4}$. Then there exist constants $c_2, c_2' > 0$ (depending only on $\theta$) such that: if $N > \frac{c_2 n^8 \kappa^8}{\rho^4}\log n$, then with probability at least $1 - n^{-c_2'}$,

$$\Big\|\frac{1}{\theta n N} \sum_{i=1}^{N} C_{x_i}^\top C_{x_i} - I\Big\| \leq \frac{\theta(1-3\theta)\rho^2}{200n^4\kappa^4}. \tag{26}$$

Next, we bound $\|C_f R - C_f(C_f^\top C_f)^{-1/2}\|$ by following the proofs of [8, Lemma 15] and [9, Lemma B.2]. Define $Q := \frac{1}{\theta n N} \sum_{i=1}^{N} C_{x_i}^\top C_{x_i}$.

$$\|C_f R - C_f(C_f^\top C_f)^{-1/2}\|$$
$$= \|C_f(C_f^\top Q C_f)^{-1/2} - C_f(C_f^\top C_f)^{-1/2}\|$$
$$\leq \sigma_1(C_f) \cdot \|(C_f^\top Q C_f)^{-1/2} - (C_f^\top C_f)^{-1/2}\|$$
$$\leq \sigma_1(C_f) \frac{\|(C_f^\top Q C_f)^{-1} - (C_f^\top C_f)^{-1}\|}{\sigma_n\big((C_f^\top C_f)^{-1/2}\big)} \tag{27}$$
$$= \sigma_1^2(C_f) \|(C_f^\top Q C_f)^{-1} - (C_f^\top C_f)^{-1}\|$$
$$\leq \frac{\sigma_1^2(C_f)}{\sigma_n^2(C_f)} \|(C_f^\top C_f)(C_f^\top Q C_f)^{-1} - I\|$$
$$= \kappa^2 \left\| \left[ I + (C_f^\top (Q - I) C_f)(C_f^\top C_f)^{-1} \right]^{-1} - I \right\|$$
$$\leq \kappa^2 \frac{\|C_f^\top (Q - I) C_f\| \|(C_f^\top C_f)^{-1}\|}{1 - \|C_f^\top (Q - I) C_f\| \|(C_f^\top C_f)^{-1}\|} \tag{28}$$
$$\leq \kappa^4 \frac{\|Q - I\|}{1 - 1/2} \leq \frac{\theta(1 - 3\theta)\rho^2}{100 n^4}. \tag{29}$$

The inequality (27) follows from the fact that, for positive definite $A$ and $B$,

$$\|A^{-1/2} - B^{-1/2}\| \leq \frac{\|A^{-1} - B^{-1}\|}{\sigma_n(A^{-1/2} + B^{-1/2})} \leq \frac{\|A^{-1} - B^{-1}\|}{\sigma_n(B^{-1/2})},$$

which in turn follows from the identity

$$(A^{-1/2} - B^{-1/2})(A^{-1/2} + B^{-1/2}) + (A^{-1/2} + B^{-1/2})(A^{-1/2} - B^{-1/2}) = 2(A^{-1} - B^{-1}).$$

The inequality (28) is due to the fact that $\|(I + A)^{-1} - I\| \leq \|(I + A)^{-1}\| \|A\| \leq \frac{\|A\|}{1 - \|A\|}$ for $\|A\| < 1$. The last line (29) follows from (26) and $\|C_f^\top (Q - I) C_f\| \|(C_f^\top C_f)^{-1}\| \leq \kappa^2 \|Q - I\| < \frac{1}{2}$.

The rest of Lemma 1.4 follows from the Lipschitz continuity of the objective function. Define $U := C_f R$, and $U' := C_f(C_f^\top C_f)^{-1/2}$, which is an orthogonal matrix. We have

$$\|C_f R\| = \|U\| \leq \|U'\| + \|U - U'\| < 2. \tag{30}$$

Recall that for the Bernoulli-Rademacher model, $\|x_i\| \leq \sqrt{n}$ and $\|C_{x_i}\| \leq n$. Then the difference of the gradients of $L(h) = \frac{1}{N} \sum_{i=1}^{N} \phi(C_{x_i} U h)$ and $L'(h) = \frac{1}{N} \sum_{i=1}^{N} \phi(C_{x_i} U' h)$ can be bounded as follows:

$$\|\nabla_L(h) - \nabla_{L'}(h)\|$$
$$\leq \max_{i \in [n]} \|U^\top C_{x_i}^\top \nabla_\phi(C_{x_i} U h) - U'^\top C_{x_i}^\top \nabla_\phi(C_{x_i} U' h)\|$$
$$\leq \max_{i \in [n]} \|U^\top C_{x_i}^\top \nabla_\phi(C_{x_i} U h) - U^\top C_{x_i}^\top \nabla_\phi(C_{x_i} U' h)\|$$
$$\quad + \max_{i \in [n]} \|U^\top C_{x_i}^\top \nabla_\phi(C_{x_i} U' h) - U'^\top C_{x_i}^\top \nabla_\phi(C_{x_i} U' h)\|$$
$$\leq \max_{i \in [n]} \|U\| \|C_{x_i}\| \cdot \sqrt{n}[3(\|U\| \|x_i\|)^2] \cdot \|U - U'\| \|x_i\|$$
$$\quad + \max_{i \in [n]} \|U - U'\| \|C_{x_i}\| \cdot \sqrt{n} \|x_i\|^3$$
$$\leq 25\sqrt{n} \cdot \max_{i \in [n]} \|C_{x_i}\| \|x_i\|^3 \|U - U'\|$$
$$\leq 25 n^3 \|U - U'\|,$$

where the third inequality follows from the fact that $\nabla_\phi(\cdot)$ is Lipschitz continous and bounded on compact sets – the Lipschitz constant of $(\cdot)^3$ on the interval $[-\|U\| \|x_i\|)^2, \|U\| \|x_i\|)^2]$ is $3(\|U\| \|x_i\|)^2$,

and the upper bound of $|(\cdot)^3|$ on the interval $[-\|x_i\|, \|x_i\|]$ is $\|x_i\|^3$. Similarly the difference of the Hessians can be bounded as follows:

$$\|H_L(h) - H_{L'}(h)\|$$

$$\leq \max_{i\in[n]} \|U^\top C_{x_i}^\top H_\phi(C_{x_i}Uh)C_{x_i}U - U'^\top C_{x_i}^\top H_\phi(C_{x_i}U'h)C_{x_i}U'\|$$

$$\leq \max_{i\in[n]} \|U^\top C_{x_i}^\top H_\phi(C_{x_i}Uh)C_{x_i}U - U^\top C_{x_i}^\top H_\phi(C_{x_i}U'h)C_{x_i}U\|$$

$$+ \max_{i\in[n]} \|U^\top C_{x_i}^\top H_\phi(C_{x_i}U'h)C_{x_i}U - U'^\top C_{x_i}^\top H_\phi(C_{x_i}U'h)C_{x_i}U\|$$

$$+ \max_{i\in[n]} \|U'^\top C_{x_i}^\top H_\phi(C_{x_i}U'h)C_{x_i}U - U'^\top C_{x_i}^\top H_\phi(C_{x_i}U'h)C_{x_i}U'\|$$

$$\leq \max_{i\in[n]} \|U\|^2\|C_{x_i}\|^2 \cdot [6(\|U\|\|x_i\|)] \cdot \|U - U'\|\|x_i\|$$

$$+ \max_{i\in[n]} \|U - U'\|\|U\|\|C_{x_i}\|^2 \cdot [3\|x_i\|^2]$$

$$+ \max_{i\in[n]} \|U - U'\|\|C_{x_i}\|^2 \cdot [3\|x_i\|^2]$$

$$\leq 57 \cdot \max_{i\in[n]} \|C_{x_i}\|^2\|x_i\|^2\|U - U'\|$$

$$\leq 57n^3\|U - U'\|,$$

where the third inequality uses the Lipschitz constant and upper bound of $3(\cdot)^2$.

It follows from (29) and the above bounds that

$$\sup_{h\in S^{n-1}} \|\widehat{\nabla}_L(h) - \widehat{\nabla}_{L'}(h)\|$$

$$\leq \sup_{h\in S^{n-1}} \|\nabla_L(h) - \nabla_{L'}(h)\|$$

$$\leq 25n^3\|U - U'\| \leq \frac{\theta(1 - 3\theta)\rho^2}{4n}.$$

$$\sup_{h\in S^{n-1}} \|\widehat{H}_L(h) - \widehat{H}_{L'}(h)\|$$

$$\leq \sup_{h\in S^{n-1}} \|H_L(h) - H_{L'}(h)\| + \sup_{h\in S^{n-1}} \|\nabla_L(h) - \nabla_{L'}(h)\|$$

$$\leq 100n^3\|U - U'\| \leq \frac{\theta(1 - 3\theta)\rho^2}{n}.$$

$\square$

*Proof of Lemma 1.5.* The set $\mathcal{H}_1''$ equals the union of $(\rho, 1)$-neighborhoods of $\{\pm e_j\}_{j=1}^n$, and the columns of $C_f^{-1} = C_g$ are the shifted versions of the inverse filter $g$. Therefore, by (14), every point $h^* \in (C_f^\top C_f)^{1/2}C_f^{-1}\mathcal{H}_1''$ satisfies

$$\|h^* \pm (C_f^\top C_f)^{1/2}\mathcal{S}_j(g)\| \leq \sqrt{2\rho},$$

for some $j \in [n]$. It follows that

$$\|Rh^* \pm \mathcal{S}_j(g)\|$$

$$\leq \|Rh^* - (C_f^\top C_f)^{-1/2}h^*\| + \|(C_f^\top C_f)^{-1/2}h^* \pm \mathcal{S}_j(g)\|$$

$$\leq \|R - (C_f^\top C_f)^{-1/2}\| + \|(C_f^\top C_f)^{-1/2}\|\|h^* \pm (C_f^\top C_f)^{1/2}\mathcal{S}_j(g)\|$$

$$\leq \frac{\theta(1 - 3\theta)\rho^2}{100n^4\sigma_n(C_f)} + \frac{\sqrt{2\rho}}{\sigma_n(C_f)}$$

$$\leq \frac{2\sqrt{\rho}}{\sigma_n(C_f)},$$

where the second to last line follows from (29), and the last line follows from $\theta(1-3\theta)\rho^2/(100n^4) < (2-\sqrt{2})\sqrt{\rho}$. Therefore,

$$\|C_f Rh^* \pm e_j\| = \|C_f(Rh^* \pm \mathcal{S}_j(g))\| \leq \|C_f\|\|Rh^* \pm \mathcal{S}_j(g)\|$$

$$\leq \sigma_1(C_f) \cdot \frac{2\sqrt{\rho}}{\sigma_n(C_f)} = 2\kappa\sqrt{\rho}.$$

$\square$

## D Proofs of Lemmas 2.1 – 2.3

*Proof of Lemma 2.1.* Clearly, $L(h) \leq 0$ for all $h \in S^{n-1}$. For the Bernoulli-Rademacher model in (A1), we have $\|x_i\| \leq \sqrt{n}$ and $\|C_{x_i}\| \leq n$. Therefore,

$$\phi(C_{y_i}Rh) = -\frac{1}{4}\|C_{x_i}C_f Rh\|_4^4$$

$$\geq -\frac{n}{4}(\|x_i\|\|C_f Rh\|)^4$$

$$\geq -4n^3,$$

where the first inequality follows from the Cauchy-Schwarz inequality, and the second inequality follows from $\|C_f Rh\| \leq \|C_f R\| \leq 2$ (see (30)). Then $L(h) = \frac{1}{N}\sum_{i=1}^{N} L_i \geq -4n^3$.

We can bound the the norm of $\nabla_L(h)$ and $H_L(h)$ similarly. To bound $\|\nabla_L(h)\|$, we observe that

$$\left|(C_{x_i}^\top \nabla_\phi(C_{y_i}Rh))_{(j)}\right| \leq \|x_i\|\|\nabla_\phi(C_{y_i}Rh)\|$$

$$\leq \|x_i\| \times \sqrt{n}(\|x_i\|\|C_f Rh\|)^3$$

$$\leq \sqrt{n}\|x_i\|^4\|C_f R\|^3$$

$$\leq 8n^2\sqrt{n},$$

and hence

$$\|\nabla_L(h)\| = \|\frac{1}{N}\sum_{i=1}^{N} R^\top C_{y_i}^\top \nabla_\phi(C_{y_i}Rh)\|$$

$$\leq \|R^\top C_f^\top\|\|\frac{1}{N}\sum_{i=1}^{N} C_{x_i}^\top \nabla_\phi(C_{y_i}Rh)\|$$

$$\leq \|R^\top C_f^\top\| \times \sqrt{n}\max_{i\in[N],\,j\in[n]}\left|(C_{x_i}^\top \nabla_\phi(C_{y_i}Rh))_{(j)}\right|$$

$$\leq 16n^3.$$

To bound $\|H_L(h)\|$, we have

$$\left|(C_{x_i}^\top H_\phi(C_{y_i}Rh)C_{x_i})_{(jk)}\right| \leq \|x_i\|^2\|H_\phi(C_{y_i}Rh)\|$$

$$\leq \|x_i\|^2 \times 3(\|x_i\|\|C_f Rh\|)^2$$

$$\leq 3\|x_i\|^4\|C_f R\|^2$$

$$\leq 12n^2,$$

and hence

$$\|H_L(h)\| = \|\frac{1}{N}\sum_{i=1}^{N} R^\top C_{y_i}^\top H_\phi(C_{y_i}Rh)C_{y_i}R\|$$

$$\leq \|R^\top C_f^\top\|\|\frac{1}{N}\sum_{i=1}^{N} C_{x_i}^\top H_\phi(C_{y_i}Rh)C_{x_i}\|\|C_f R\|$$

$$\leq \|C_f R\|^2 \times n\max_{i\in[N],\,j\in[n],\,k\in[n]}\left|(C_{x_i}^\top H_\phi(C_{y_i}Rh)C_{x_i})_{(jk)}\right|$$

$$\leq 48n^3.$$

$\square$

*Proof of Lemma 2.2.* For $z \perp h$, and $h' = \frac{h+z}{\|h+z\|} = \frac{h+z}{\sqrt{1+\|z\|^2}}$, $\widehat{L}(z) = L(h')$, $\widehat{L}(0) = L(h)$, and

$\nabla_{\widehat{L}}(0) = \widehat{\nabla}_L(h)$. By the mean value theorem, there exists a convex combination $h''$ of $h$ and $h'$ such that $L(h') - L(h) = \langle h' - h, \nabla_L(h'') \rangle$, and a convex combination of $h'''$ of $h$ and $h''$ such that $\nabla_L(h'') - \nabla_L(h) = H_L(h''')(h'' - h)$. It follows that

$$\left| L(h') - L(h) - \langle z, \widehat{\nabla}_L(h) \rangle \right|$$
$$= \left| \langle h' - h, \nabla_L(h'') \rangle - \langle z, \nabla_L(h) \rangle \right|$$
$$\leq \left| \langle h' - h - z, \nabla_L(h'') \rangle \right| + \left| \langle z, \nabla_L(h'') - \nabla_L(h) \rangle \right|$$
$$\leq \frac{\|z\|^2}{1 + \sqrt{1 + \|z\|^2}} \|\nabla_L(h'')\| + \|z\| \|H_L(h''')\| \|h'' - h\|$$
$$\leq \frac{\|z\|^2}{2} \times 16n^3 + 48n^3 \|z\| \|h - h'\|$$
$$\leq 64n^3 \|z\|^2,$$

where the third inequality follows from Lemma 2.1, and the last inequality follows from the fact that $\|h - h'\| \leq \|z\|$. □

*Proof of Lemma 2.3.* Suppose the columns of orthogonal matrices $V, V' \in \mathbb{R}^{n \times (n-1)}$ form bases for the tangent spaces at $h$ and $h'$. Then a matrix representation of $D\mathcal{A}(h)$ in (12) as a mapping from the tangent space of $h$ to the tangent space at $h'$ is $V'^{\top} V (I_{n-1} - \gamma V^{\top} \widehat{H}_L(h) V)$.

Since $\langle h, h' \rangle = \|h\|^2 = 1$, we have $|\det(V'^{\top} V)| = \langle h, h'/\|h'\| \rangle = 1/\|h'\| > 0$.

By Lemma 2.1, for all $h \in S^{n-1}$,

$$\|\widehat{H}_L(h)\| \leq \|H_L(h)\| + \|\nabla_L(h)\|$$
$$\leq 48n^3 + 16n^3 = 64n^3.$$

Therefore $I_{n-1} - \gamma V^{\top} \widehat{H}_L(h) V$ is strictly positive definite for $\gamma < 1/(64n^3)$.

It follows that

$$|\det(D\mathcal{A}(h))| = |\det(V'^{\top} V)| \cdot |\det(I_{n-1} - \gamma V^{\top} H_L(h) V)| > 0.$$

□

## Footnotes

[1]For an example of finding sparse null vectors using truncated power iteration, we refer the readers to our previous paper [5, Section II].