[Reviews · NeurIPS 2018]

Reviewer 1



Summary: The authors study the geometry of multichannel sparse blind deconvolution problems over the unit sphere. The geometric analysis presented reveals two important results: (1) all local minima of the sparsity promoting objective function correspond to signed shifted versions of the true solution; (2) The objective function is strongly convex in neighborhoods of the local minima, and has strictly negative curvature directions in neighborhoods of local maxima and saddle points. Assessment: This is an excellent paper, which will be of great interest to the NIPS audience. It is well motivated, clearly written, and well organized. The work is properly placed in the context of prior work. The technical contribution is significant and original and brings great insight about an important class of problems.

Reviewer 2



This paper studies the optimization landscape of multichannel sparse blind deconvolution, and uncovers it has the desirable “all local minima are global” and “strict saddle properties“ that have been shared to a few other problems such as dictionary learning and low-rank matrix estimation. This is an important message and allows the use of manifold optimization algorithms to solve the nonconvex problem in a global manner. Further comments: -the sample complexity of the analysis seems to be very high compared to what was indicated possible from the simulations. it will be useful if the authors can comment on why there is such a big gap in the sample complexity requirement; -can the Bernoulli-Radamacher model be extended to Bernoulli-subGaussian model? The former is quite restrictive and rarely holds in practice. -though first-order methods are desirable as the per-iteration cost is low, Theorem 4.1 has a rather slow convergence rate. In particular, the algorithm should converge geometrically in the vicinity of the ground truth. It may still be beneficial to use second-order methods to reduce the high number of iterations required in Theorem 4.1 for first order methods; -the proof roadmap is similar to what was proposed in the paper “the landscape of non-convex losses” by Mei et.al. The author should compare their approach with this paper and discuss the differences. -the comparisons in Figure 4 seem quite unfair and misleading since the latter off-the-grid algorithms do not use the fact that the signal is sparse in the Fourier basis. A more fair comparison should take all direction-of-arrivals off the grid, and then compare the performance of the algorithms. -to handle miscalibrated PSF in super-res fluorescence microscopy, it might be useful to use total least squares to partially learn the PSF.

Reviewer 3



This paper considered multichannel sparse blind deconvolution(MSBD) and proposed to solve a nonconvex optimization problem for recovering the underlying signal and filters. Under some technical assumptions, the authors showed that all local minima of the objective function correspond to the inverse filter of the ground truth signal up to an inherent sign and shift ambiguity, and all saddle points have strictly negative curvatures. Finally, the authors proposed to solve this nonconvex optimization problem via a Riemannian gradient descend method. My main comments are as follows. 1) The authors did not clarify why they use the nonconvex optimization in (P1), what is the main idea behind this, and intuitively why this formulation will lead to correct recovery. 2) Compared to the reference [48], the difference is that this paper considered ‘multichannel’ (i.e., a bunch of channels (#N) to convolve with the ground truth signal and measure a bunch of (#N) observations). Despite this difference, these two papers share similarities: the objective is the same (except an additional summation appeared in this paper) and the main purpose is to analyze the geometry of the corresponding nonconvex optimization problem. However, it is unclear what are the technical difficulties and challenges for the multichannel case. It could be just a direct extension of the result in [48] and thus it is hard to assess the value of contribution and significance of this paper. 3) The Bernoulli-Rademacher model for the channels is too strong and is far away to real-world applications. The authors should clarify why using this assumption for the channels. %%%%%% after rebuttal %%%%%% I thank the authors for the response to my concerns. It is very good to see that all the results can be extended to Bernoulli-subGaussian model. The Bernoulli-subGaussian model would make the results more general and applicable. So I suggest to at least add discussion about this in the final version. Also the explanation of the equivalence between maximizing \ell_4^4 and minimizing l0 norm could be in more detail. It is not very clear to see this relationship from the plot of \ell_4 norm.